EMBO
Molecular Medicine

# SOX9 predicts progression toward cirrhosis in patients while its loss protects against liver fibrosis

Varinder S Athwal[1,2,†], James Pritchett[3,†], Jessica Llewellyn[1], Katherine Martin[1,2], Elizabeth Camacho[4], Sayyid MA Raza[1,2], Alexander Phythian-Adams[5], Lindsay J Birchall[1,2], Aoibheann F Mullan[1,2], Kim Su[1,2], Laurence Pearmain[1,2], Grace Dolman[6], Abed M Zaitoun[7], Scott L Friedman[8], Andrew MacDonald[5] (iD), William L Irving[6,9], Indra N Guha[6], Neil A Hanley[1,2] & Karen Piper Hanley[1,2,*] (iD)

## Abstract

Fibrosis and organ failure is a common endpoint for many chronic liver diseases. Much is known about the upstream inflammatory mechanisms provoking fibrosis and downstream potential for tissue remodeling. However, less is known about the transcriptional regulation *in vivo* governing fibrotic matrix deposition by liver myofibroblasts. This gap in understanding has hampered molecular predictions of disease severity and clinical progression and restricted targets for antifibrotic drug development. In this study, we show the prevalence of SOX9 in biopsies from patients with chronic liver disease correlated with fibrosis severity and accurately predicted disease progression toward cirrhosis. Inactivation of *Sox9* in mice protected against both parenchymal and biliary fibrosis, and improved liver function and ameliorated chronic inflammation. SOX9 was downstream of mechanosignaling factor, YAP1. These data demonstrate a role for SOX9 in liver fibrosis and open the way for the transcription factor and its dependent pathways as new diagnostic, prognostic, and therapeutic targets in patients with liver fibrosis.

**Keywords** extracellular matrix; hepatic stellate cells; liver fibrosis; SOX9; YAP1
**Subject Categories** Digestive System; Metabolism

## Introduction

In response to inflammation, the secretion of collagen-rich pathological extracellular matrix (ECM) by liver myofibroblasts, called hepatic stellate cells (HSCs), destroys normal tissue architecture and causes fibrosis. Fibrosis occurs in both hepatic parenchymal and biliary diseases and is a major cause of morbidity and mortality against which there are no approved drug treatments (Iredale, 2007; Friedman, 2008; Lee & Friedman, 2011; Lee *et al*, 2015). Even after withdrawal of the inflammatory insult, fibrosis can progress unchecked to organ failure and death. Alternatively, in some individuals liver fibrosis may not advance despite persisting insult (Ellis & Mann, 2012; Kisseleva *et al*, 2012; Troeger *et al*, 2012). Addressing this uncertainty requires *in vivo* knowledge of the molecular mechanisms that regulate fibrosis.

The transcription factor sex determining region Y box 9 (SOX9) is responsible *in vivo* for the organized deposition of collagen as part of cartilage formation. In humans, mutations in the *SOX9* gene cause the haploinsufficiency disorder campomelic dysplasia (CD), characterized by failed chondrogenesis (Foster *et al*, 1994; Wagner *et al*, 1994; Pritchett *et al*, 2011). In the liver, SOX9 is ordinarily expressed in cholangiocytes lining the bile ducts and plays roles in bile duct development and in hepatic regeneration (Antoniou *et al*, 2009; Carpentier *et al*, 2011; Yanger *et al*, 2013; Tarlow *et al*, 2014; Font-Burgada *et al*, 2015; Jors *et al*, 2015; Lu *et al*, 2015). *In vitro*, we have shown that SOX9 becomes expressed during the activation of HSCs by profibrotic signaling factors when it promotes

1 Division of Diabetes, Endocrinology and Gastroenterology, Faculty of Biology, Medicine & Health, Manchester Academic Health Science Centre, University of Manchester, Manchester, UK
2 Research & Innovation Division, Central Manchester University Hospitals NHS Foundation Trust, Manchester, UK
3 School of Healthcare Science, Manchester Metropolitan University, Manchester, UK
4 Centre for Health Economics, Institute of Population Health, Faculty of Medical & Human Sciences, Manchester Academic Health Science Centre, University of Manchester, Manchester, UK
5 Manchester Centre for Collaborative Inflammation Research, Faculty of Life Sciences, University of Manchester, Manchester, UK
6 Nottingham Digestive Diseases Centre and National Institute for Health Research (NIHR) Nottingham Biomedical Research Centre, Nottingham University Hospitals NHS Trust and University of Nottingham, Nottingham, UK
7 Department of Cellular Pathology, Nottingham Digestive Diseases Centre and National Institute of Health Research Biomedical Research Unit in Gastroenterology and Liver Disease, University of Nottingham and Nottingham University Hospitals NHS Trust, Nottingham, UK
8 Division of Liver Diseases, Icahn School of Medicine at Mount Sinai, New York, NY, USA
9 School of Life Sciences, Nottingham Digestive Diseases Centre and National Institute of Health Research Biomedical Research Unit in Gastroenterology and Liver Disease, University of Nottingham and Nottingham University Hospitals NHS Trust, Nottingham, UK
*Corresponding author. Tel: +44 161 3060643; E-mail: karen.piperhanley@manchester.ac.uk
†These authors contributed equally to this work

production of ECM components, such as type 1 collagen (COL1) and osteopontin (OPN) (Hanley *et al*, 2008; Pritchett *et al*, 2012). However, these studies relied on imperfect *in vitro* cell models. Direct *in vivo* evidence for SOX9 function in liver fibrosis (rather than regeneration) has been lacking.

In this study, we have studied SOX9 in patients with chronic liver disease where sequential liver biopsies enabled fibrosis severity and disease progression to be deciphered over time and examined the consequences of inactivating SOX9 in rodent models of parenchymal and biliary liver fibrosis.

# Results

### SOX9 profile in the liver of patients with chronic liver disease

Patients chronically infected with hepatitis C virus (HCV) show variable severity of liver fibrosis with the potential for marked disease progression within a few years. While these patients are now routinely treated with antiviral drugs, this clinical setting offered a distinctive opportunity to study SOX9 and fibrosis progression because previous routine surveillance entailed patients undergoing accurately staged serial biopsies from their liver. We accessed the UK Trent HCV Cohort and identified 115 patients with initial and follow-up biopsy categorized for severity by a histopathologist using the 7-point Ishak (IS) fibrosis stage, IS0 to IS6; the latter representing the most severe fibrosis/cirrhosis (Ishak *et al*, 1995) (Table EV1). 70% of patients had no or mild fibrosis on initial biopsy (IS0-1; $n = 80$) rather than more intermediate (IS2-3; 16%, $n = 19$) or severe disease (IS4-6; 14%, $n = 16$). However, at follow-up biopsy 36.9 $\pm$ 2.5 months later, this percentage declined to 54% ($n = 62$) with corresponding increases in both the IS2-3 category (24%, $n = 28$) and severe (22%, $n = 25$) fibrosis. Thus, over 3 years liver fibrosis had progressed in 30 patients (26%), regressed in eight patients (7%), and remained unchanged in 77 (67%) allowing us to correlate SOX9 with severity (tissue remained from 152 biopsies for this purpose; Fig EV1) and disease progression (where tissue remained from 59 paired biopsies; Fig EV1).

Staining for collagen using picrosirius red (PSR) and αSMA (a marker of activated myofibroblasts) was increased with higher Ishak fibrosis stage particularly in regions of scarring (Fig 1A and B). In IS0 (no liver fibrosis), SOX9 was observed in the nuclei of cholangiocytes (Fig 1B; Antoniou *et al*, 2009; Rowe *et al*, 2013). In moderate and severe fibrosis, nuclear SOX9 was progressively more noticeable in cells with elongated nuclei and αSMA. SOX9 was also detected at slightly lower levels in cells with large round nuclei adjacent to the scar costained for the hepatocyte marker, α1-antitrypsin (α1AT), consistent with regenerative hepatocytes (termed Hep from here on) (Fig 1B and C). SOX9 was present in all $CK19^+$ and $CK7^+$ cholangiocytes lining bile ducts; these cells were not detected outside of duct structures. SOX9 was not detected in the $CD34^+$ vasculature. Given the role of SOX9 in regenerative hepatocytes (Kawaguchi, 2013), we studied EpCAM which marks cholangiocytes and nascent hepatocytes (Yoon *et al*, 2011) and compared this to the $SOX9^+$ ductal plate in human fetal liver, which gives rise to periportal hepatocytes and biliary cells (Carpentier *et al*, 2011; Furuyama *et al*, 2011; Rowe *et al*, 2013). In moderate and severe fibrosis, EpCAM localized to the $SOX9^+$ Heps adjacent to the scar in

an arrangement remarkably similar to the fetal ductal plate (Fig 1D). Thus, SOX9 could be partitioned discretely into bile duct nuclei and post-injury expression in isolated cells demarcating a putative HSC/Hep population.

### SOX9 index correlates with severity of liver fibrosis in patients and predicts progression

We used the above profiles to construct a SOX9 index of positive nuclei per surface area of biopsy. We counted the biliary, HSC/Hep, and total $SOX9^+$ populations in 152 biopsies. While there was no change in $SOX9^+$ biliary cells at different stages of fibrosis, the HSC/Hep SOX9 index doubled as fibrosis severity increased; a finding which was also reflected in the total count (Fig 1E). HSC/Hep and total SOX9 indices from the initial biopsies correlated very strongly with the increment in Ishak fibrosis stage between initial and follow-up biopsies, while the biliary count did not alter (Fig 1F). The HSC/Hep SOX9 index was progressively higher for those patients whose liver fibrosis worsened over the subsequent 3 years. To see whether we could detect mild disease that was destined to progress, we identified all patients whose initial biopsy was IS0-2 and classified them into two groups: "progressors" ($n = 12$), defined by a worsening fibrosis at follow-up biopsy of $\geq 2$ Ishak stages; or "non-progressors" ($n = 25$) where the fibrosis either improved was static or worsened by no more than one stage. Patients across the two groups had not received anti-HCV treatment, showed no significant difference in age, gender, alcohol consumption, ethnicity, serum alanine aminotransferase (ALT), necroinflammation, or HCV genotype (Appendix Table S1). In contrast, the HSC/Hep SOX9 index was significantly increased, more than double, in the initial biopsy of the "progressors" compared to "nonprogressors" ($P < 0.01$; Fig 1G). The potential to predict future progression was analyzed by calculating the area under the receiver operating characteristic curve (AUROC). The ideal test would have an AUROC of 1, where a random guess would have an AUROC of 0.5. The HSC/Hep SOX9 index (AUROC = 0.895, $P < 0.001$), and consequently the total SOX9 index (AUROC = 0.910, $P < 0.05$), forecast significant worsening of fibrosis amongst patients in the "progressor" category (Fig 1H). Moreover, when the initial biopsy was subdivided into individual Ishak stages (rather than an amalgamated IS0-2) or into gender, the total SOX9 index significantly predicted progression in each instance except for patients at IS2 where low numbers were limiting (Appendix Table S2). We determined positive and negative predictive values (PPV and NPV) around the mean SOX9 count for total, HSC/Hep, and biliary populations (Fig 1I–K). The HSC/Hep and total SOX9 indices were predictive (Fig 1I and J), while the biliary count was unhelpful (Fig 1K). For additional comparison, we also quantified necroinflammation (NI) using the 0–18 scale of minimal (1–3), mild (4–8), moderate (9–12), and severe (13–18) NI (Ishak *et al*, 1995). The mean NI score in the initial biopsy was 2.75. Applying this threshold gave a PPV of 67% (NPV 63%). Setting the threshold at the upper limit of mild NI (score = 8) excluded all except one patient (Fig 1L; i.e., less predictive than the HSC/Hep and total SOX9 indices).

Univariate logistic regression analysis identified age, the HSC/Hep (or total) SOX9 count, and NI to be significant risk factors in disease progression (Table 1; $P < 0.05$). However, when adjusted for time between biopsies using multivariate analysis, only age and

Figure 1.

**Figure 1. SOX9 in human liver fibrosis.**

A    Liver biopsies showing no (IS0), moderate (IS3), and severe (IS6) fibrosis stained by PSR. Size bar, 300 μm.

B    Immunohistochemistry (brown staining) for αSMA and SOX9 on consecutive sections from patients with no (IS0), moderate (IS3), and severe (IS6) fibrosis. Boxed areas are shown to the right at higher magnification. Bile duct (bd). SOX9 detected in HSCs with elongated nuclei (hatched line) next to scar and in hepatocytes with rounded nuclei (arrows). Size bars, 200 μm.

C    Immunofluorescence for SOX9 (red) costained with CK19, CK7, α1AT, and αSMA (green). Arrowheads show elongated nuclei in αSMA-positive HSCs.

D    Immunohistochemistry for SOX9 and EpCAM (brown staining) on consecutive sections from severe (IS6) fibrosis and normal fetal liver at 18 weeks post-conception (wpc). Dual immunofluorescence confirms SOX9 and EpCAM in the same cell in severe fibrosis.

E–G    Data for SOX9 index for the biliary, HSC/Hep, and total (biliary + HSC/Hep) populations. In worsening categories of liver fibrosis (E). In the initial biopsy plotted against the increment in Ishak fibrosis stage between the initial and follow-up biopsy (F). The lines show the best fit by ordinal logistic regression analysis. In the initial biopsy categorized by whether patients did or did not progress by at least two stages on follow-up biopsy (G).

H    Receiver operator curves for the sensitivity and specificity of the SOX9 index in the initial biopsy as a predictor of non-progressors and progressors.

I–L    Plots of SOX9 index (total, HSC/Hep, and biliary) and NI score marking progressors in green and non-progressors in red. The solid line indicates the mean in each graph. The broken line in NI dot plot (L) indicates the threshold (score = 8) below which all NI is categorized as "mild".

Data information: Unless otherwise indicated in text, two-tailed unpaired *t*-test was used for statistical analysis. Data in bar charts show means ± s.e.m. *P*-values are indicated. Sample numbers for (E and F) (*n* = 152); (G), non-progressors (*n* = 25) and progressors (*n* = 12).

the SOX9 index retained significance. The odds ratios indicated that each 1-year increase in age was associated with a 4% increase in the risk of becoming a "progressor", whereas each cell increase in either the HSC/Hep or total SOX9 index increased the risk of fibrosis progression by 12% (Table 1). Male gender, serum ALT levels, or NI at first biopsy did not associate significantly with fibrosis progression.

**Sox9 is induced in HSCs by liver injury and its loss protects against fibrosis and chronic inflammation**

To investigate the mechanism underlying SOX9 function in liver fibrosis, we explored the consequence of its loss in two well-established rodent models of the disease. Liver fibrosis was induced by carbon tetrachloride ($CCl_4$) (with olive oil control) injections given twice weekly for 8 weeks to induce parenchymal disease or the bile duct ligated (BDL) to cause peribiliary fibrosis at 2 weeks. While SOX9 persisted in cholangiocytes, the nuclear transcription factor became detected in both models in a subset of $HNF4\alpha^+$ cells with large round nuclei scattered around the bile ducts, equivalent to the regenerative Heps observed in human tissue (Fig 2A and B, and Appendix Fig S1). SOX9 protein and mRNA were detected in αSMA-positive spindle-shaped cells in sections of liver fibrosis induced by $CCl_4$ and BDL (Fig 2C) as previously (Hanley *et al*, 2008; Pritchett *et al*, 2012). To corroborate that these cells were indeed activated

HSCs, we isolated HSCs following acute liver injury with $CCl_4$ (i.e., before scarring interferes with the HSC isolation procedure) and showed without culture that SOX9 was increased compared to control (Fig 2D and Appendix Fig S2). This *ex vivo* isolation from fibrotic liver has led to these cells being termed "*in vivo* activated" HSCs (De Minicis *et al*, 2007; Mederacke *et al*, 2015; Martin *et al*, 2016). Taken together, these data were entirely consistent with our observations in fibrotic human liver (Fig 1).

*Sox9* null mice develop severe developmental defects and die in the perinatal period (Pritchett *et al*, 2011). Therefore, we used tamoxifen (Tam) to remove SOX9 expression in Sox9$^{fl/fl}$;ROSAC-reER$^{+/-}$ adult mice with Sox9$^{fl/fl}$;ROSACreER$^{-}$ animals as control. Tamoxifen was injected twice in the week prior to fibrosis induction in both BDL and $CCl_4$ models and every 2 weeks during the 8-week regime of $CCl_4$ injury. Tamoxifen effectively excised the *Sox9* gene and removed SOX9 protein (Fig 2E and Appendix Fig S3). Removal of SOX9 in adult mice did not compromise survival (corroborated by others Mori-Akiyama *et al*, 2007). Tamoxifen did not induce ectopic SOX9 expression in uninjured control livers (Appendix Fig S3) or alter SOX9 detection in the HSC/Hep and biliary compartments of injured control livers. Knocking out *Sox9* did not alter the liver/body weight ratio in uninjured or fibrotic settings (Appendix Fig S4). In both parenchymal and peribiliary models, fibrosis was clearly improved by loss of *Sox9* with reduced

**Table 1. Logistical regression analysis indicating risk factor association with progression of liver fibrosis.**

| Predictor | Odds ratio (95% CI) | | |
| --- | --- | --- | --- |
| | **Unadjusted** | **Adjusted for age** | **Adjusted for time to progression** |
| Age, per year increase | 1.08 (1.00, 1.17)* | – | 1.04 (1.00, 1.08)* |
| Age, > 55 versus ≤ 55 years | 4.36 (0.36, 53.39) | – | – |
| Gender, male versus female | 3.09 (0.56, 17.17) | 3.32 (0.50, 21.92) | 2.87 (0.44, 18.70) |
| Total Sox9, per unit increase | 1.13 (1.04, 1.23)* | 1.13 (1.03, 1.23)* | 1.12 (1.03, 1.22)* |
| Sox9 isolated cells, per unit increase | 1.13 (1.04, 1.23)* | 1.14 (1.04, 1.25)* | 1.12 (1.03, 1.23)* |
| Sox9 BD, per unit increase | 1.14 (1.00, 1.29)* | 1.14 (0.99, 1.31) | 1.09 (0.95, 1.25) |
| ALT, per unit increase | 1.01 (0.99, 1.02) | 1.01 (0.99, 1.02) | 1.01 (0.99, 1.02) |
| NI, per unit increase | 1.63 (0.98, 2.71)* | 1.55 (0.90, 2.68) | 1.68 (0.94, 3.01) |

*Significant association (*P* < 0.05).

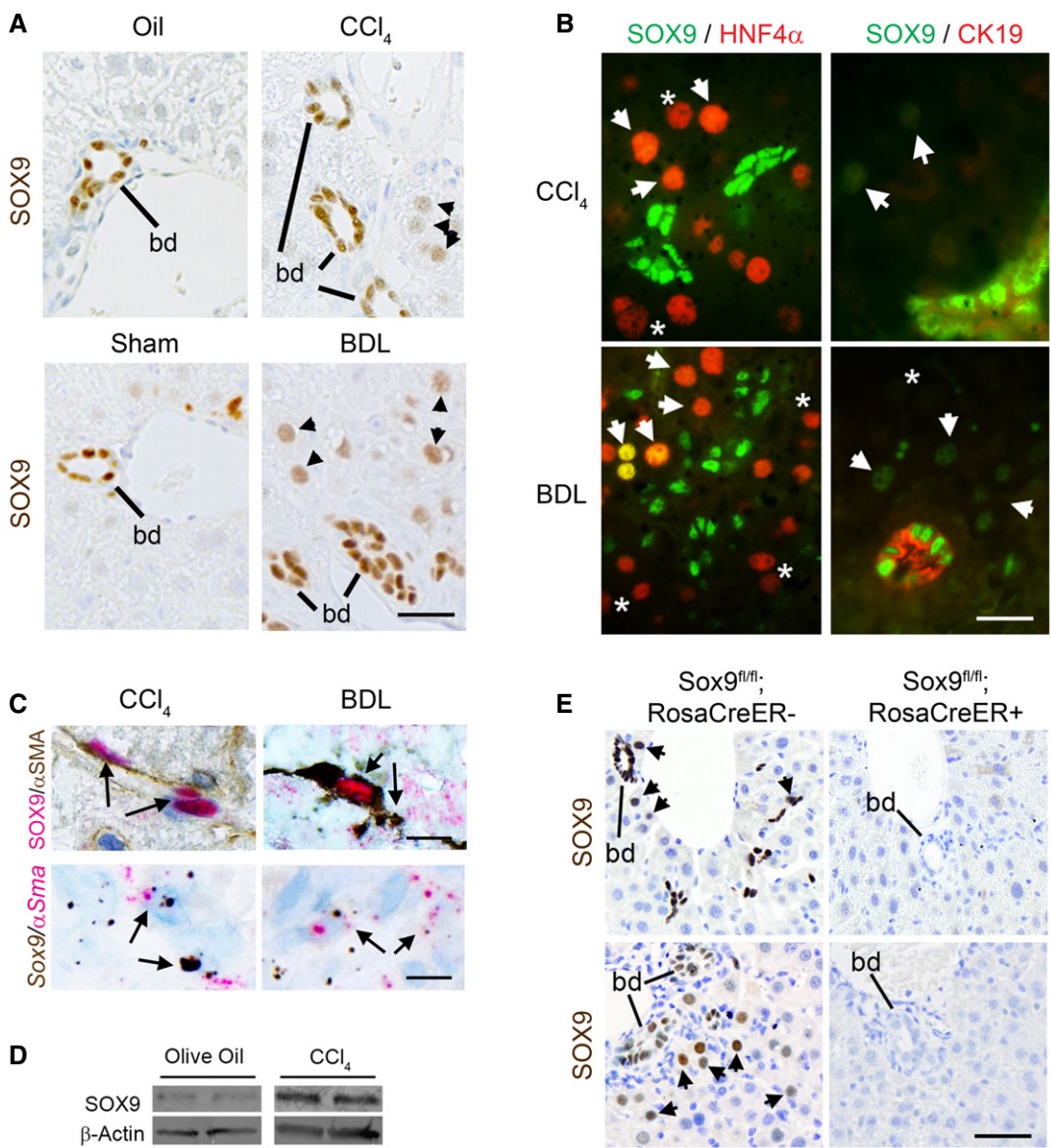

**Figure 2. SOX9 is involved in liver fibrosis *in vivo*.**

A   Immunohistochemistry for SOX9 (brown) in wild-type (WT) mice following CCl₄- or BDL-induced fibrosis. SOX9-positive bile ducts (bd) and hepatocytes (arrowheads) indicated.

B   (Left panel) Immunofluorescence for SOX9 (green) and HNF4α (red) in WT fibrotic mice. Arrowheads (orange/yellow staining) indicate SOX9⁺/HNF4α⁺ hepatocytes, and star (*; red) indicates SOX⁻/HNF4α⁺ hepatocytes. (Right panel) Immunofluorescence for SOX9 (green nucleus) and CK19 (red cytoplasm) in WT fibrotic mice. Arrowheads (green) indicate SOX9⁺ hepatocytes, and star (*) indicates SOX⁻ hepatocytes identified by DAPI staining in Appendix Fig S1.

C   Co-localization by immunohistochemistry (top panel) and *in situ* hybridization (lower panel) for SOX9 (red) and αSMA (brown) in wild-type (WT) mice following CCl₄- or BDL-induced fibrosis.

D   Expression of SOX9 protein by immunoblotting of *in vivo* activated HSCs extracted from WT mice following CCl₄ injections compared to olive oil control.

E   Immunohistochemistry for SOX9 (brown) in control (Sox9^{fl/fl};RosaCreER⁻) or Sox9-null (Sox9^{fl/fl};RosaCreER⁺) mice following fibrosis. Bile ducts (bd) and hepatocytes (arrowheads) indicated. SOX9 is absent, even in bile ducts in Sox9-null liver.

Data information: Size bar, 25 μm (A, B and E), 10 μm (C).
Source data are available online for this figure.

deposition of collagen by PSR staining and less surface area covered by αSMA staining (Fig 3A–F). Liver function, assessed by serum biochemistry, was also markedly improved, especially in the BDL model in which serum ALT and bilirubin were normalized (Fig 3G and H). In patients with parenchymal liver injury, scars which

extend from one portal tract to another are termed "bridging fibrosis" and, when numerous, are a marker of severity (Ishak fibrosis score ≥ 4 on the IS0-6 7-point scale) (Ishak *et al*, 1995). Loss of SOX9 reduced the incidence of bridging fibrosis induced by CCl₄ injury by at least one-third (Fig 3I), while in BDL, bile duct

**Figure 3.  Loss of Sox9 improves liver fibrosis *in vivo*.**

Representative images and quantification shown for olive oil-treated (*n* = 6) or chronic CCl₄ (*n* = 5 Sox9^fl/fl;RosaCreER⁻; *n* = 8 Sox9^fl/fl;RosaCreER⁺)-induced fibrosis or following sham operation (*n* = 5) or BDL (*n* = 7 Sox9^fl/fl;RosaCreER⁻; *n* = 5 Sox9^fl/fl;RosaCreER⁺))-induced fibrosis.

A    Picrosirius red (PSR) staining (collagen deposition in red) counterstained with fast green (top row) and immunohistochemistry for α-SMA (brown staining bottom row; activated HSC/myofibroblast marker) in olive oil-treated (left) or chronic CCl₄-induced fibrosis (right) in control and Sox9-null mice. Size bars = 200 μm.

B, C    Quantification of surface area covered by the PSR staining or α-SMA in control (Cnt) and Sox9-null (Null) in (A).

D    PSR staining (red; top row) and immunohistochemistry for α-SMA (brown; middle row) and CK19 (brown; bottom row) in control and Sox9-null mice following Sham operation (left) or BDL-induced fibrosis (right). Size bars = 500 μm.

E, F    Quantification of surface area covered by the PSR staining or α-SMA in (D).

G, H    Liver function is improved in Sox9-null mice following CCl₄ (G)- or BDL (H)-induced fibrosis compared to control mice. Reduction in serum alanine aminotransferase (ALT; G, H) and bilirubin (H) down to levels shown in non-fibrotic mice (olive oil-treated groups; oil).

I    Sox9 loss improved severity of fibrosis compared to control mice indicated by quantification of bridging fibrosis in PSR sections following CCl₄. Olive oil-treated mice had no bridging fibrosis, in line with a non-fibrotic liver histology in (A).

J    Ductal hyperplasia as quantified by the surface area covered by CK19-positive ducts in (D) is reduced in Sox9-null mice compared to control following BDL.

Data information: All mice were treated with tamoxifen (Tam) which did not induce ectopic expression of SOX9 in non-fibrotic livers (Fig 2A and E, and Appendix Fig S3). All experiments are *n* ≥ 5 as indicated. Two-tailed unpaired *t*-test was used for statistical analysis. Data in bar charts show means ± s.e.m. *P*-values indicated.

hyperplasia (assessed by surface area immunostained for CK19) was halved (Fig 3J). In both CCl4 and BDL models, tissue architecture was improved by H&E histology staining and, in keeping with our previous work (Hanley et al, 2008; Pritchett et al, 2012, 2014; Martin et al, 2016), profibrotic COL1 expression was greatly reduced in response to Sox9 loss in vivo and in vitro (Appendix Figs S5–S7).

Given the critical developmental role for SOX9 in multiple tissues (Pritchett et al, 2011), it was not possible to knock out SOX9 in HSCs using the currently published PdgfrB, AP2, and LRAT Cre models (Henderson et al, 2013; Mederacke et al, 2013; Moran-Salvador et al, 2013). PdgfrB expression overlaps SOX9 in neural crest cell (NCC) populations responsible for heart development (Akiyama et al, 2004; Van den Akker et al, 2008; Smith & Tallquist, 2010). In keeping with this, > 80% of our Sox9$^{fl/fl}$;PdgfrBCre$^+$ mice died at birth. AP2 also has a major role in NCCs (and chondrogenesis) (Wenke & Bosserhoff, 2010; Pritchett et al, 2011). LRAT expression overlaps SOX9 during development [e.g., in liver, lung, pancreas (Batten et al, 2004; Jennings et al, 2013; Pritchett et al, 2011)]. Thus, none of these Cre drivers were suitable for HSC-specific SOX9 inactivation. In light of this, to determine which of the three SOX9$^+$ liver cell types (cholangiocyte, HSC, and/or Hep) harbored the transcription factor's profibrotic function we took advantage of an AlbuminCre (AlbCre) allele to generate Sox9$^{fl/fl}$;AlbCre$^{+/-}$ mice, which excised Sox9 from hepatocytes and cholangiocytes (both lineages develop from Alb$^+$ progenitors) but not HSCs (Fig 4A and B, and Appendix Figs S8 and S9). Uninjured adult animals were healthy with normal hepatic and biliary function as expected (Poncy et al, 2015). Fibrotic ECM deposition in Sox9$^{fl/fl}$;AlbCre$^{+/-}$ mice was as extensive as in control mice in both chronic CCl4 and BDL models (Fig 4C–E). The liver/body weight ratio was unaltered (Appendix Fig S10). Surface area covered by αSMA staining was no different (Appendix Fig S11). SOX9 protein remained in discrete elongated cells in Sox9$^{fl/fl}$;AlbCre$^{+/-}$ mice, and transcript was localized to αSma$^+$ cells (Fig 4A and B, and Appendix Fig S8). To further demonstrate retention of SOX9 in the HSC lineage, in vitro activation of HSCs from Sox9$^{fl/fl}$;AlbCre$^{+/-}$ livers showed normal induction of SOX9 protein (Fig 4F and G, and Appendix Fig S9). From these combined data in Sox9$^{fl/fl}$;ROSACreER$^{+/-}$ and Sox9$^{fl/fl}$;AlbCre$^{+/-}$ mice, we conclude that SOX9's major profibrotic function is in HSCs.

Next, we explored the effect of eliminating SOX9 expression on chronic inflammation in liver fibrosis. In mice, there is a natural tendency for liver fibrosis to resolve after cessation of CCl4 injury reliant on a phenotypic switch in monocyte-derived liver macrophages from high (profibrotic) to low ("restorative") Ly6C expression (Ramachandran et al, 2012). This observation was maximal 72-h post-CCl4 injection amongst an F4/80$^+$ cell population (Ramachandran et al, 2012). To probe this further, we isolated the monocyte–macrophage lineage (Ly6C$^+$CD64$^+$) from the wider pool of CD45$^+$CD11B$^+$ myeloid cells 72 h after conclusion of a 4-week program of biweekly CCl4 injections (Appendix Fig S12). The fibrotic livers from control and Sox9 null mice contained four distinct populations according to levels of Ly6C and MHCII, a specific macrophage marker (Fig 5A and B, and Appendix Fig S12). Populations 1 and 2 with intermediate or high Ly6C levels lacked macrophage maturity markers (MerTK, CD64, and F4/80) consistent with infiltrating monocytes (Fig 5A and B). Population 3 (MHCII$^{+-}$ Ly6C$^{Hi/Int}$) contained a proportion of cells positive for MerTK,

CD64, and F4/80, while virtually all population 4 (the Ly6C$^{Low}$ cells) expressed high levels of the macrophage maturity markers (Fig 5A and B). In fibrotic livers from control mice, sorting by F4/80$^+$ identified more Ly6C$^{Low}$ than Ly6C$^{High/int}$ cells consistent with previous work (Ramachandran et al, 2012; Fig 5C). However, by MHCII sorting, the proportion of Ly6C$^{Hi/+}$ and Ly6C$^{Int/-}$ cells were equivalent (Fig 5D and E). Against this backdrop, we scrutinized chronic inflammation in the tamoxifen-treated Sox9$^{fl/fl}$;ROSACreER$^{+/-}$ mice. The size of the monocyte–macrophage infiltrate in the liver was similar [i.e., the sum of populations 1–4 was unchanged (Fig 5D and E)]. However, populations 1 and 2 were expanded, while populations 3 and 4 were diminished (visible in Fig 5A and B and quantified in Fig 5D and E). The intensity of F4/80 immunostaining was also reduced in the absence of SOX9 (Fig 5F). As SOX9 is not expressed in the hematopoietic lineage, the implications of these data are twofold: that monocyte recruitment following injury is unaffected; however, in the context of SOX9-dependent reduced scarring and improved liver function, maturation of macrophages is blocked.

## Mechanosensitive signaling regulates SOX9 via YAP1 during liver fibrosis

We have previously shown SOX9 is induced by the key profibrotic signaling factors transforming growth factor beta (TGF-β) and hedgehog (HH). To explore this further, we reasoned that upregulated SOX9, and consequently increased ECM deposition, in fibrotic liver might relate to tissue stiffness. Therefore, we extracted HSCs and cultured them on acrylamide hydrogels to model normal liver elasticity or the fibrotic environment at 4 or 12 kPa, respectively. Yes-associated protein-1 (YAP1) is a mechanosensitive transcription factor that when dephosphorylated is nuclear and active with co-factors, transcriptional coactivator with PDZ-binding motif (TAZ, also known as WWTR1) and TEA domain factor (TEAD). In contrast, when phosphorylated, YAP1 is cytoplasmic and inactive. In HSCs cultured on 4 kPa hydrogels, YAP1 was cytoplasmic, SOX9 not detected, and αSMA, TAZ, and TEAD only weakly present (Fig 6A and B). In stark contrast, under the 12 kPa conditions modeling fibrosis, YAP1, TAZ, and TEAD were robustly detected in the nucleus with cytoplasmic αSMA (Fig 6A and B). Moreover, when YAP1:TEAD complexes were disrupted using their specific inhibitor, verteporfin, SOX9 protein levels were reduced by 60% in activated HSCs (Fig 6C). Previous studies by us and others have shown that verteporfin improves aspects of liver fibrosis in vivo (Mannaerts et al, 2015; Martin et al, 2016). In the setting of verteporfin treatment, SOX9 counts were reduced in the HSC/Hep compartment in both CCl4-induced liver injury and BDL (Fig 6E and F).

## Discussion

SOX9 is a key transcription factor regulating multiple components of the ECM during normal development and in studies modeling isolated HSC function in vitro (Hanley et al, 2008; Pritchett et al, 2012). Here, we studied the requirement for SOX9 in vivo in liver fibrosis. Several conclusions can be drawn. Firstly, loss of SOX9 in vivo abrogated fibrosis in response to liver injury. SOX9 expression was restricted to cholangiocytes, and post-injury HSCs and

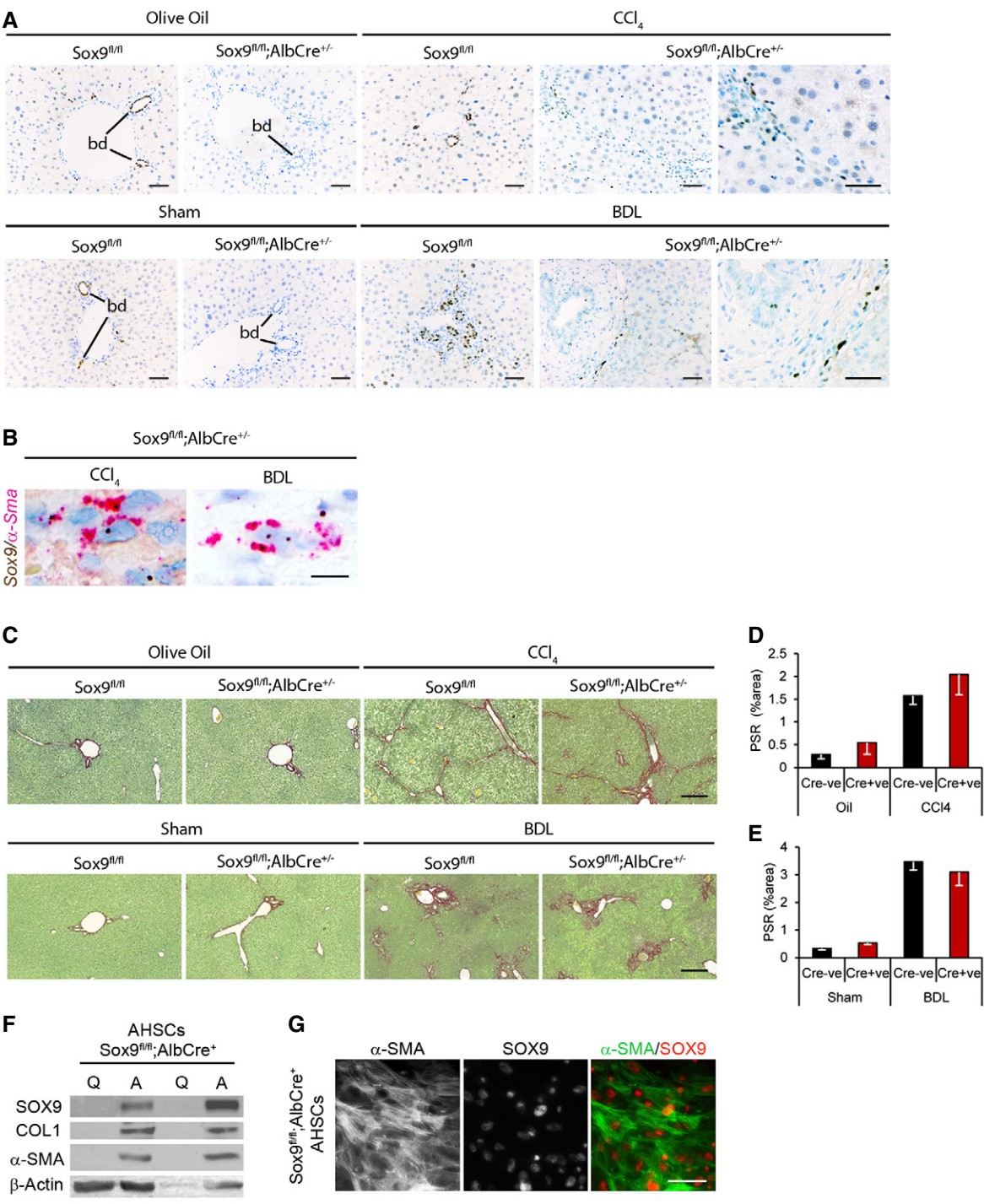

**Figure 4. Fibrosis in Sox9^{fl/fl};AlbCre^{+/−} mice.**

A–C Representative images shown for olive oil-treated ($n = 5$) or chronic CCl$_4$ ($n = 5$)-induced fibrosis or following sham operation ($n = 6$ Sox9$^{fl/fl}$;AlbCre$^{−}$; $n = 5$ Sox9$^{fl/fl}$; AlbCre$^{+}$) or BDL ($n = 5$ Sox9$^{fl/fl}$;AlbCre$^{−}$; $n = 8$ Sox9$^{fl/fl}$;AlbCre$^{+}$)-induced fibrosis. SOX9 immunohistochemistry (brown; A), *in situ* hybridization for *Sox9* (brown) and *α-Sma* (red) and collagen deposition by PSR staining (red; B) in control (Sox9$^{fl/fl}$;AlbCre$^{−}$) or Sox9-null (Sox9$^{fl/fl}$;AlbCre$^{+}$) mice following fibrosis. Higher magnified image of SOX9 localization in discrete cells within the scar is shown for CCl$_4$ and BDL in the Sox9$^{fl/fl}$;AlbCre$^{+}$ mice (A).

D, E Quantification of surface area covered by the PSR staining in (C).

F Expression of SOX9 protein by immunohistochemistry in quiescent (Q) and activated (A) HSCs extracted from Sox9$^{fl/fl}$;AlbCre$^{+}$ mice.

G Individual fluorescent channels showing localization and expression of α-SMA (left panel), SOX9 (middle panel), and composite image for α-SMA (green) and SOX9 (red) in right panel in activated Sox9$^{fl/fl}$;AlbCre$^{+}$ HSCs (genotyping shown in Appendix Fig S9).

Data information: Two-tailed unpaired *t*-test was used for statistical analysis. Data in bar charts show means ± s.e.m. Size bars = 100 μm (A), 10 μm (B) 200 μm (C), 25 μm (G).

Source data are available online for this figure.

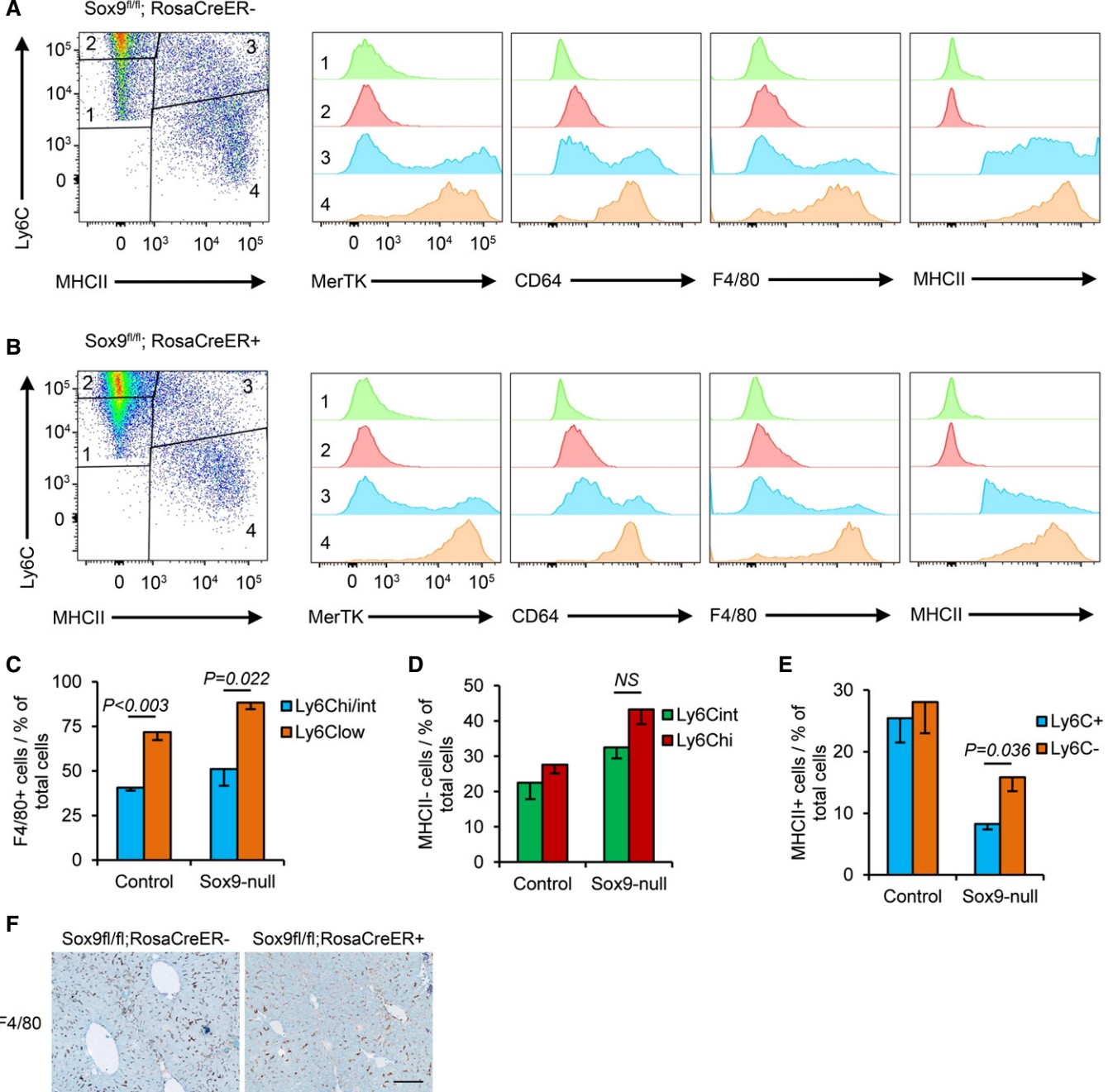

**Figure 5. Ly6C^low pro-resolution macrophages predominate in Sox9-null animals.**

A, B   Flow cytometry of CD45+ CD11B+ cells in control (Sox9^fl/fl;RosaCre⁻; A) and Sox9-null (Sox9^fl/fl;RosaCre⁺; B) livers stained for Ly6C and MHCII (far left) to identify four populations of cells with differences in their expression of MerTK, CD64, F4/80, and MHCII.

C   Graphical representation of Ly6C and F4/80 staining in macrophages from populations 3 and 4.

D, E   Graphical representation of cell numbers in the four Ly6C and MHCII cell populations in control and Sox9-null livers. Data are representative of four independent experiments.

F   F4/80 immunohistochemistry (brown staining) in control and Sox9-null mice livers following 4-week CCl$_4$-induced fibrosis. Size bar = 100 μm.

Data information: Two-tailed unpaired $t$-test was used for statistical analysis. Data in bar charts show means ± s.e.m. $P$-values indicated. All experiments are $n$ = 4.

some hepatocytes (most likely regenerative). There was no impact from eliminating SOX9 in hepatocytes and cholangiocytes [which furthers the argument against epithelial-to-mesenchymal transition as a significant mechanism for liver fibrosis (Scholten

*et al*, 2010; Taura *et al*, 2010; Wells, 2010)]. Our data point toward the HSC as the causative cell type-mediating fibrosis and that this lineage emerges separately from albumin-expressing hepatobiliary precursors (Yin *et al*, 2013). Unaltered biliary hyperplasia in our

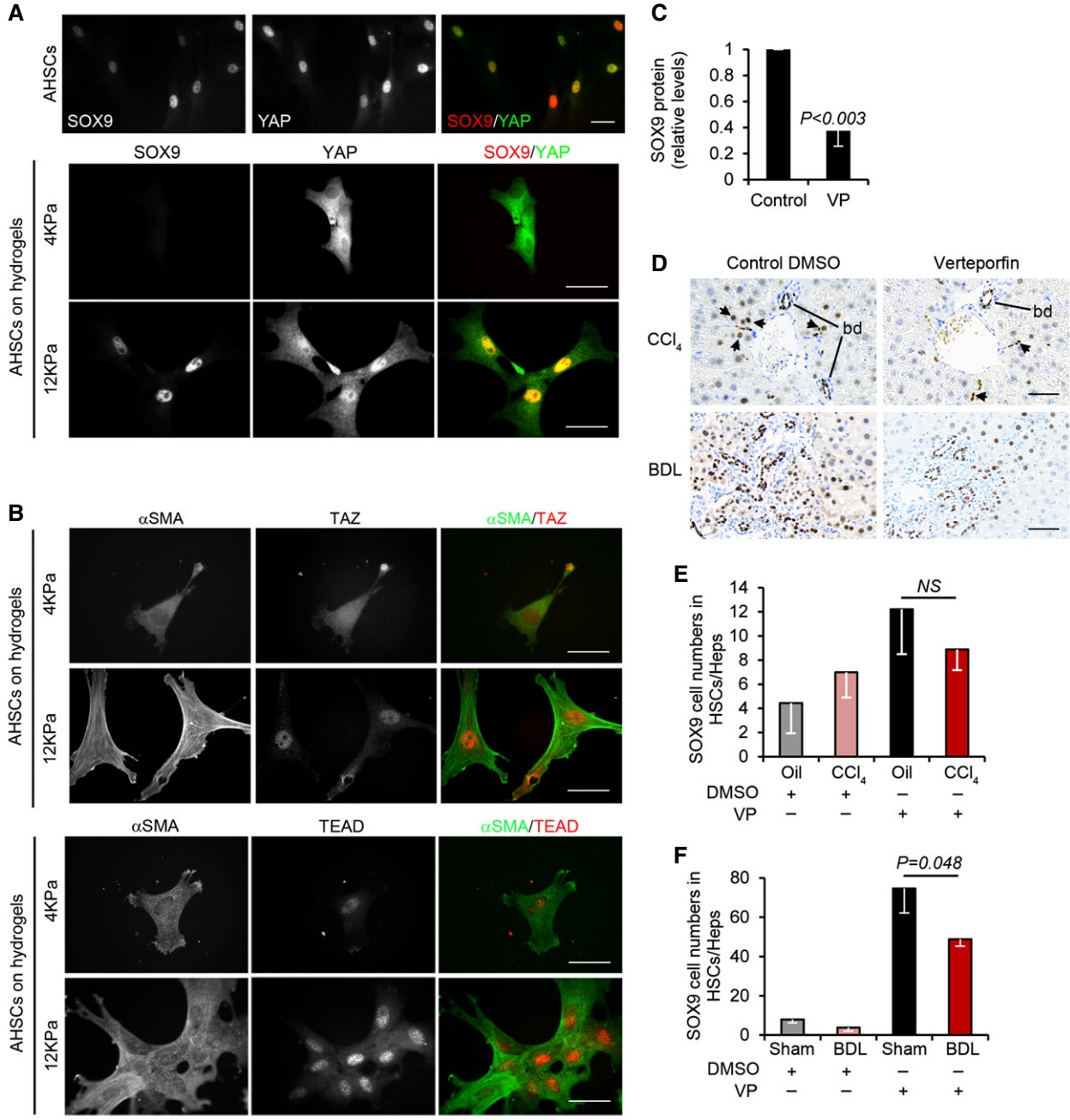

**Figure 6.  YAP signaling regulates SOX9.**

A, B   Immunofluorescence of activated mouse HSCs cultured on plastic (A) or hydrogels (A, B). Individual fluorescent channels showing localization and expression of SOX9 (left), YAP (middle), and composite image for SOX9 (red) and YAP (green) in right panel. (B) Individual and composite images for αSMA (green) and TAZ (red) or TEAD (red). Size bars, 25 μm.

C      Quantified decrease in SOX9 protein levels following inhibition of YAP using verteporfin (VP) in activated mouse HSCs expressed relative to DMSO control (n = 3).

D      Immunohistochemistry for SOX9 (brown) following CCl$_4$- or BDL-induced fibrosis in mice treated with DMSO (CCl$_4$ n = 4, BDL n = 7) or verteporfin (CCl$_4$ n = 3, BDL n = 5). Size bar, 50 μm.

E, F   Quantification of SOX9 cell numbers in the HSC/Hep population.

Data information: Two-tailed unpaired *t*-test was used for statistical analysis. Data in bar charts show means ± s.e.m. *P*-values are indicated.

injured Sox9$^{fl/fl}$;AlbCre$^{+/-}$ mice also implies that SOX9 is not an intrinsic requirement for post-injury proliferation of cholangiocytes.

To translate, the functional requirement for SOX9 in fibrosis from mouse to human is challenging. Genetic evidence in human

would require inactivating mutations or haploinsufficiency protecting against liver fibrosis, but campomelic dysplasia is a multisystem disorder lethal around birth (Foster *et al*, 1994; Wagner *et al*, 1994). Activating mutations or duplications of the

*SOX9* locus (a cause of 46,XX disorder of sex development) that might promote liver fibrosis are exceptionally rare and lack the additional context of liver injury. Our data on a cohort of patients with serial biopsies over time and variable fibrosis progression (Ghany *et al*, 2009; Liang & Ghany, 2013) provide three strands of evidence that SOX9's role in liver fibrosis in mouse seems likely in human: The profile of SOX9 detection following liver injury in patients was identical to that observed in mouse; the prevalence of *de novo* SOX9 expression in biopsies correlated precisely with fibrosis severity; and thirdly, the extent of SOX9 at early stages of disease predicted progression within 3 years. Indeed, the SOX9 index performed better than other fibrosis risk factors and could potentially assist the stratification of patients with liver fibrosis alongside other measures, such as the cirrhosis risk score (Huang *et al*, 2007; Marcolongo *et al*, 2009; Trepo *et al*, 2011).

Taken together, our data support a model whereby injury induces SOX9 downstream of YAP1 in HSCs to promote liver fibrosis. Our data open up the potential for SOX9 and its dependent pathways as biomarkers or as targets for lessening liver scarring, improving hepatic chronic inflammation and halting the progression of liver fibrosis toward cirrhosis.

## Materials and Methods

### Study design

The goal of this study was to provide comprehensive evidence *in vivo* in mouse and human that the transcription factor SOX9 and its associated pathways promote liver fibrosis and its progression toward cirrhosis. All experiments were blinded to the person analyzing the data. For animal experiments, where possible, the minimum number of animals per fibrosis protocol was five based on accepting an 80% chance of detecting a 30% change in collagen content from normal versus fibrotic animals at the level of $P \geq 0.05$. However, where additional animals have been available, we have included these in our analysis to improve power and our ability to statistically detect smaller physiological effects.

For all human studies, informed consent was obtained from all subjects (including embryonic and fetal tissue) and experiments conformed to the principals set out in the WMA Declaration of Helsinki and the Department of Health and Human Services Belmont Report.

### Animal experiments

All mouse strains were maintained on a C57BL/6J background in a 12-h light–dark cycle with water and food provided *ad libitum*. Sox9$^{fl/fl}$ mice were a kind gift from Professor Gerd Scherer (Kist *et al*, 2002). RosaCreER (Seibler *et al*, 2003) and AlbuminCre (Postic *et al*, 1999) were purchased from Jackson Laboratories. To allow inducible global *Sox9* deletion, Sox9$^{fl/fl}$ mice were crossed with RosaCreER mice to generate Sox9$^{fl/fl}$;RosaCreER animals. To delete *Sox9* in cholangiocytes and hepatocytes, Sox9$^{fl/fl}$ mice were crossed with AlbCre mice to give Sox9$^{fl/fl}$;AlbCre$^{+/-}$ animals. REDExtract-N-AMP™ Tissue PCR kit (Sigma, UK) was used to genotype DNA prepared from ear clips. Three

primers were used to detect Sox9 floxed alleles and recombination events (Reverse: CGCTGGTATTCAGGGAGGTACA; F1: CTCCGGTAG CAAAGGCGTTTAG; F2: CCGGCTGCTGGGAAAGTATATG). Reverse and F1 gave a 314-bp product only if recombination had occurred. Reverse and F2 gave either a 247-bp product (wild type) or a 419-bp (floxed allele) product (Kist *et al*, 2002). For RosaCreER genotyping, forward (CCATCATCGAAGCTTCACTGAAG) and reverse (GGAGTTT CAATACCCGAGATCATGC) primers gave a 310-bp product. To keep the possibility of spontaneous Cre activity in the absence of tamoxifen to a minimum, only RosaCreER heterozygote animals were used. To allow detection of the wild-type Rosa26 allele and to confirm animals were heterozygous wild type, forward (CATGTCTTTAATCTACCTC GATGG) and reverse primers (CTCTTCCCTCGTGATCTGCAACTCC) were used to give a 299-bp product. To delete *Sox9* in cholangiocytes and hepatocytes, Sox9$^{fl/fl}$ mice were crossed with AlbCre mice to give Sox9$^{fl/fl}$:AlbCre$^{+/-}$ animals. Only heterozygous animals were used, and genotype was confirmed using three primers (common Cre reverse: TCGTTGCATCGACCGGTAAT; Albumin gene reverse: TAGCATGGTCGAGCAGGCAG; Albumin promoter forward: AGAGC GAGTCTTTCTGCACACAG) which gives a 430-bp product in the presence of the wild-type albumin allele and a 200-bp product in the presence of the AlbCre allele.

Animals were housed and maintained and animal experiments performed under approval from the University of Manchester Ethical Review Committee and in accordance with UK Government Home Office license for animal research. Carbontetrachloride (CCl$_4$) or bile duct ligation (BDL) was used to induce chronic liver fibrosis in age-matched male mice (between 8 and 14 weeks old at the start of experiments), and YAP signaling was interrogated during fibrosis using verteporfin (VP) (Sigma, UK) as previously described (Martin *et al*, 2016). All randomly assigned experimental and controls were littermate sex- and age-matched mice.

For CCl$_4$ fibrosis was induced by 2-µl intraperitoneal (i.p.) injections of sterile CCl$_4$ (Sigma, UK) per g body weight in a ratio of 1:3 by volume in olive oil, or olive oil alone (control) twice weekly for 8 weeks. Tissue and serum were collected at the end of the experiments. For BDL-induced fibrosis, under anesthesia two ligatures were placed around the exposed bile duct. Sham (control) animals underwent the same procedure without ligation of the bile duct. To improve post-operative survival, animals were kept in a thermoneutral warm cabinet (30°C) following BDL, and given soaked diet. Tissue and serum were collected 14 days after surgery.

To induce activation of Cre and global deletion of *Sox9* in Sox9$^{fl/fl}$;RosaCreER animals, tamoxifen (Sigma, UK) was given by i.p injection. To control for any unexpected effects of tamoxifen, both Sox9$^{fl/fl}$;RosaCreER$^{+/-}$ and Sox9$^{fl/fl}$;RosaCreER$^{-/-}$ animals were injected. For CCl$_4$, 4 mg tamoxifen (20 mg/ml in olive oil) was injected 8 and 7 days before the first CCl$_4$ injection (Day 0). Subsequent 2 mg doses were given fortnightly, starting on day 6, for the duration of the experiment. For BDL, 4 mg tamoxifen (20 mg/ml in olive oil) was injected i.p. 8 and 7 days prior to surgery. 2 mg was injected i.p. 7 days after surgery.

For both CCl$_4$ and BDL models of fibrosis in Sox9$^{fl/fl}$;RosaCreER animals, there were four experimental groups. CCl$_4$: Sox9$^{fl/fl}$;RosaCreER$^{-/-}$ Olive Oil ($n = 6$); Sox9$^{fl/fl}$;RosaCreER$^{+/-}$ Olive Oil ($n = 6$), Sox9$^{fl/fl}$;RosaCreER$^{-/-}$ CCl$_4$ ($n = 5$), and Sox9$^{fl/fl}$;RosaCreER$^{+/-}$ CCl$_4$ ($n = 8$). BDL: Sox9$^{fl/fl}$;RosaCreER$^{-/-}$ Control ($n = 5$);

Sox9$^{fl/fl}$;RosaCreER$^{+/-}$ Control ($n = 5$), Sox9$^{fl/fl}$;RosaCreER$^{-/-}$ BDL ($n = 7$), and Sox9$^{fl/fl}$;RosaCreER$^{+/-}$ BDL ($n = 5$). To determine the effects of deleting *Sox9* in both cholangiocytes and hepatocytes, CCl$_4$ or BDL were used to induce liver fibrosis in Sox9$^{fl/fl}$;AlbCre$^{-/-}$ and Sox9$^{fl/fl}$;AlbCre$^{+/-}$ mice. For both models of fibrosis, this gave four experimental groups. CCl$_4$: Sox9$^{fl/fl}$;AlbCre$^{-/-}$ olive oil ($n = 5$); Sox9$^{fl/fl}$;AlbCre$^{+/-}$ olive oil ($n = 5$), Sox9$^{fl/fl}$;AlbCre$^{-/-}$ CCl$_4$ ($n = 5$) and Sox9$^{fl/fl}$;AlbCre$^{+/-}$ CCl$_4$ ($n = 5$). BDL: Sox9$^{fl/fl}$;AlbCre$^{-/-}$ sham ($n = 6$); Sox9$^{fl/fl}$;AlbCre$^{+/-}$ sham ($n = 5$), Sox9$^{fl/fl}$;AlbCre$^{-/-}$ BDL ($n = 5$) and Sox9$^{fl/fl}$;AlbCre$^{+/-}$ BDL ($n = 8$). A minimum of 5 mice were used per group. Recombination status and genotype were confirmed by PCR of DNA extracted from liver tissue.

To interrogate YAP signaling during fibrosis, *in vivo* verteporfin (VP) (Sigma, UK) was injected by i.p. (Martin *et al*, 2016). Wild-type *C57BL/6J* mice (Charles River, UK) were given i.p. CCl$_4$ or olive oil vehicle control as described above [and described previously (Martin *et al*, 2016)] for a period of 6 weeks; or bile duct ligated. 10 μl/g body weight of 10 mg/ml VP in 10% DMSO in PBS was injected i.p. three times weekly during the last 3 weeks of CCl$_4$ treatment (on alternate days to CCl4 or olive oil), or every 48 h starting on day 7 after BDL, to give a dose of 100 mg/kg body weight. Control mice were injected with 10 μl/g body weight of 10% DMSO in PBS. This gave four treatment groups for each model of liver fibrosis. CCl$_4$: olive oil with DMSO ($n = 5$); olive oil with VP ($n = 3$), CCl4 with DMSO ($n = 3$); and CCl4 with VP ($n = 4$). BDL: sham with DMSO ($n = 8$), sham with VP ($n = 4$), BDL with DMSO ($n = 7$), BDL with VP ($n = 5$). Levels of fibrosis and liver function serum tests have been previously described for these animals (Martin *et al*, 2016).

Liver tissue from all *in vivo* experiments was fixed in 4% PFA and processed for histology as previously reported (Martin *et al*, 2016).

### Primary cell extraction and culture

Primary rat and mouse (Male Sprague Dawley, Charles River, UK) hepatic stellate cells (rHSCs/mHSCs, respectively) or mouse *in vivo* activated HSCs were isolated and cultured as previously described (Hanley *et al*, 2008; Pritchett *et al*, 2012; Martin *et al*, 2016). To interrogate YAP1 signaling, verteporfin (VP) (Sigma, UK) was used to disrupt YAP1:TEAD complexes (Liu-Chittenden *et al*, 2012). HSCs were cultured for 8 days, treated with 10 μM VP or vehicle control for 24 h and then harvested for protein or qPCR analysis. To model changes in mechanical stiffness, HSCs were seeded on gels of varying stiffness (4 or 12 kPa, Cell Guidance Systems, UK). HSCs were then harvested for protein or qPCR analysis, or fixed with 4% paraformaldehyde (Pritchett *et al*, 2014) for immunofluorescence.

### Protein expression

Protein expression was analyzed as previously described (Martin *et al*, 2016). Antibodies used were raised against SOX9 (Millipore AB5535 1:5,000) and anti-rabbit HRP-conjugated secondary antibody (GE Healthcare NA934V 1:10,000). Quantity One software (Bio-Rad) was used for image acquisition and data analysis. HRP-conjugated anti-β-actin antibody (Sigma A3854 1:30,000) was used as a loading control.

### Immunohistochemistry and immunocytochemistry

Tissue samples were fixed in 4% paraformaldehyde and processed either for paraffin embedding and sectioning. Livers were embedded in identical orientation to allow direct comparison of tissue. For paraffin sections, tissue was dehydrated and embedded in paraffin. Cells cultured on chamber slides were fixed in 4% paraformaldehyde and stored at 4°C in PBS. For diaminobenzidine (DAB) labeling, only tissue sections were incubated in 3% (v/v) H$_2$O$_2$ (Sigma, UK) to quench endogenous peroxidase activity. Antigen retrieval was carried out by boiling in 10 mM sodium citrate (pH 6) for 10 min, except for EpCAM IHC, for which antigen retrieval was performed using pepsin reagent (Sigma, UK) at 37°C for 10 min. Antigen retrieval was not performed on cells grown on hydrogels. Tissue sections or cells were incubated in primary antibody in 0.1% Triton X-100 in PBS containing 3% serum (serum was from the same species the secondary antibody was raised in), overnight at 4°C. Antibodies used were anti-CK19 (Developmental Studies Hybridoma Bank 1:300), anti-CK7-FITC (Abcam AB118958 1:50), anti-EpCAM (Leica Novocastra NCL-ESA 1:200), anti-F4/80 (Abcam AB6640 1:500), anti-HNF4α (R&D Systems PP-H1415-00 1:100), anti-SOX9 (Millipore AB5535 1:2,000), anti-TAZ (Cell Signaling #4883 1:1,000), anti-TEAD (Cell Signaling #12292 1:1,000), anti-YAP1 (Santa Cruz Sc-1011199 1:200), anti-α1AT (Bethyl Labs A80-122A 1:1,000), anti-α-SMA (ICC: Leica Novocastra SMA-R-7-CE 1:100; IHC: DAKO M0851 1:100), and anti-COL1 (IHC: Southern Biotech 1310-01 1:100). For α-SMA IHC on mouse tissue sections, Mouse on Mouse basic detection kit (Vector Laboratories) and Vectastain Elite Avidin Biotin Complex (ABC) system (Vector Laboratories) were used. For bright-field αSMA/SOX9 dual, the Mouse on Mouse Impress HRP polymer kit (MP-2400) and Impress AP anti-rabbit (alkaline phosphatase) Polymer Detection kit (MP-5401)/Impact Vector Red AP substrate (SK-5105; all from Vector Laboratories) were used, respectively. After DAB (or Red AP) labeling tissue was counterstained with Toluidine Blue. For immunofluorescence, secondary antibodies used were 488 or 594 Alexa Fluors (Molecular Probes, Invitrogen) raised against the appropriate species (1:1,000).

### *In situ* mRNA hybridization

*In situ* detection of *Sox9* and *αSma* RNA transcripts was carried out on paraffin-embedded tissue sections using the RNAScope assay (Advanced Cell Diagnostics). Sections were pretreated using an extended protease treatment and hybridized under conditions as described (RNAScope Sample Preparation and Pretreatment Guide) using automated RNAScope probes for *Sox9*, *αSma*, and standard negative *Dapb* (a bacterial gene) and positive *Ppib* control probes. Detection was by RNAScope LS 2.5 Duplex red/brown Assay for the Leica Bond RX autostainer (Catalog no. 322440) and Bond Polymer Refine Red (DS9390) and Brown DAB (DS9800). Slides were counterstained with hematoxylin.

### Histology and analysis

Slides for morphological analysis were stained with hematoxylin and eosin (H & E) as previously described (Piper *et al*, 2004). Fast green picrosirius red (PSR) was used to stain liver sections for collagen (Pritchett *et al*, 2014; Martin *et al*, 2016). The extent of scarring

was determined by morphometric analysis of PSR and myofibroblast accumulation by surface area covered by α-SMA staining. The 3D Histech Panoramic 250 Flash II slide scanner was used to acquire images of all three liver lobes on each slide. At 10× magnification, 10 images were selected from each slide at random and analyzed with Adobe Photoshop. Stained pixels were selected using the Colour Range tool and expressed as a fraction of the total number of pixels. Analysis was performed blind following randomization of slides and the acquired images.

## FACS

Mononuclear blood cells (MNCs) were prepared from whole livers of ROSACreER$^{-/-}$ and ROSACreER$^{+/-}$ mice following 4 weeks' CCl$_4$ treatment as described above. Animals were perfused with HBSS (0.5 mM EDTA) via the left ventricle of the heart 72 h after the final CCl$_4$ injection. Liver tissue was incubated with 5 ml/g digestion buffer (0.8 U/ml Liberase-TM (Roche), 160 U/ml DNase I Type IV (Sigma) in HBSS) for 30 min at 37°C. After homogenization using a 100-μm cell strainer and subsequent wash steps, MNCs were purified by centrifugation in a 33% Percoll density gradient. Red blood cells were lysed using RBC Lysis buffer (Sigma). Following FCR block, cells were stained for 30 min on ice with antibodies of interest at the appropriate dilutions as determined by titration: B220-BV785 (1:100 Biolegend), CD11b-BV711 (1:450 Biolegend), CD11c-BV605 (1:200 Biolegend), CD45-AF700 (1:400 eBioscience), CD64-PE (1:100 Biolegend), F4/80-APC (1:300 eBioscience), Ly-6C-PE/Cy7 (1:500 Biolegend), Ly-6G-PerCP/eF710 (1:400 eBioscience), MerTK-Biotin (1:100 R&D Systems), MHCII-eF450 (1:500 eBioscience), Siglec-F-PE/CF594 (1:400 BD Biosciences), Streptavidin-PE/Cy5 (Biolegend). Cells were washed in PBS and analyzed (BD FACS LSR Fortessa and FlowJo software). Compensation was performed using UltraComp eBeads (eBioscience). Cells passed through the cytometer were gated by granularity and size to select single cells. Live/Dead Blue (Invitrogen) was used to select live cells. CD45$^+$ cells were gated as shown in Appendix Fig S7 to eliminate eosinophils, neutrophils, and B cells. CD64$^+$ Ly6C$^+$ monocytes and macrophages were selected from the Cd45$^+$ Cd11b$^+$ myeloid population for further analysis.

## Human embryonic and fetal tissue

Human embryonic and fetal tissue was collected, used, and stored with ethical approval from the North West Research Ethics Committee, following codes of practice given by the Human Tissue Authority under the legislation of the UK Human Tissue Act 2008. Tissue collection and handling was as previously described (Piper *et al*, 2004; Hanley *et al*, 2008; Jennings *et al*, 2013). Briefly, following medical or surgical terminations of pregnancy, human embryos and fetal tissues were collected and fixed in 4% paraformaldehyde and embedded in paraffin wax for sectioning at 5-μm intervals.

## Human liver biopsy collection and cell counting

Paired human liver biopsies were obtained with informed consent and ethical approval from the Trent Cohort Study of HCV antibody-positive patients from across the former UK Trent Health Region (Mohsen, 2001; Ryder *et al*, 2004). Patient selection and data collection have been described previously (Mohsen, 2001; Ryder *et al*,

### The paper explained

#### Problem
Liver fibrosis is a major, increasing cause of death characterized by extracellular matrix (ECM) secretion from myofibroblasts that causes scarring and, ultimately, organ failure requiring transplant. Although potentially reversible, predicting progression is limited and there are currently no approved antifibrotic therapies to treat the disease. To address, this requires *in vivo* knowledge of the core molecular mechanisms that regulate fibrosis.

#### Results
SOX9 is a key transcription factor regulating expression of multiple ECM components during normal development and in studies modeling isolated liver myofibroblast function *in vitro*. Here, we show that prevalence of SOX9 in biopsies from patients with chronic liver disease correlates with fibrosis severity and accurately predicts disease progression toward cirrhosis. Inactivation of *Sox9* in mice protects against both parenchymal and biliary fibrosis, improves liver function, and ameliorates chronic inflammation. Using multiple transgenic models to inactivate SOX9, results support hepatic myofibroblasts as the causative cell type-mediating fibrosis.

#### Impact
This study addresses an area of enormous unmet clinical need. SOX9 outperformed all other current measures predicting outcome in liver fibrosis and would allow novel patient stratification for care and treatment. Our data support a critical role for SOX9 in liver myofibroblasts to promote liver fibrosis and open up the potential for SOX9 and its dependent pathways as biomarkers or antifibrotic targets for improving liver scarring.

2004; Williams *et al*, 2009). Patients who received intervening therapy or were infected with human immunodeficiency virus (HIV) were excluded (Ryder *et al*, 2004). Liver biopsies were assessed blindly by one of three histopathologists (Ryder *et al*, 2004). Biopsy specimens > 1.5 cm long with ≥ 5 portal tracts that stained for SOX9 by IHC were digitized using a Nanozoomer 2.0-HT slide scanner and viewed with the accompanying imager (Hamamatsu, Japan). In stained biopsy sections, five portal tract regions were selected per slide at 200× magnification leading to the analysis of 0.28 mm$^2$ per portal tract, within which all SOX9-positive cells were counted and categorized by location within or outside bile ducts to provide a mean ± standard error ("SOX9 index"). The SOX9 index was determined with no prior knowledge of disease stage, whether initial or follow-up biopsy or whether fibrosis had progressed.

## Statistics

For *in vivo* experiments, two-way ANOVA followed by Bonferroni post hoc analysis was used to determine statistical significance. For FACS experiments, significance was calculated using unpaired *t*-tests with Welch's correction. Unless otherwise stated, for all other experiments statistical significance was determined by two-tailed Student's *t*-test. All experiments were carried out three times or more ($n = 3$) as indicated.

For analysis of SOX9 in human liver biopsies, statistical analysis (SPSS version 19) used either a paired two-tailed Student's *t*-test (paired data) or a chi-squared ($\chi^2$) test (categorical data). In groups with 3 or more nominal variables, means were compared by one-way analysis of variance (ANOVA) with Tukey's honest

significance test (HSD) post hoc correction. Ordinal logistical regression analysis was used to evaluate the relationship between SOX9 cell count and the progression of liver fibrosis categorized by Ishak stage with McFadden $R^2$ indicating the degree of correlation. Area under the receiver operator curve (AUROC) was used to quantify the potential of the SOX9 count to predict progression of liver fibrosis between initial and follow-up biopsy. Logistic regression (STATA version 10) examined the impact of a number of factors on the likelihood of showing disease progression. Results were reported as odds ratios (OR) with 95% confidence intervals (CI).

**Expanded View** for this article is available online.

## Acknowledgements

This work was supported by the Medical Research Council (MRC; KPH, MR/J003352/1 & MR/P023541/1; KM and VSA are MRC Clinical Training Fellows; SR is a KRUK Clinical Training Fellow); the Manchester Biomedical Research Centre and the Wellcome Trust (NAH, WT088566MA). The Core Histology Facility at the University of Manchester is acknowledged for their technical help and support. We acknowledge technical assistance from Kara Simpson.

## Author contributions

KPH and NAH conceived and designed experiments. VSA and JP contributed to experimental planning and design. KM, AMZ, AM, AP-A, GD, SLF, WLI, and ING provided reagents and contributed to experimental design. VSA, JP, JL, KM, SMAR, APA LJB, AFM, KS, and LP performed experiments. VSA, JP, JL, AP-A, and KPH analyzed the data. VSA and EC performed statistical analysis. KPH and NAH guided experiments, analyzed data, and wrote the manuscript.

## Conflict of interest

The authors declare that they have no conflict of interest.

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
