## [Review Process File · EMBO Molecular Medicine]

SOX9 predicts progression towards cirrhosis in patients while its loss protects against liver fibrosis

Varinder S Athwal, James Pritchett, Jessica Llewellyn, Katherine Martin, Elizabeth Camacho, Sayyid MA Raza, Alexander Phythian-Adams, Lindsay J Birchall, Aoibheann F Mullan, Kim Su, Laurence Pearmain, Grace Dolman, Abed M Zaitoun, Scott L Friedman, Andrew MacDonald, William Irving, Indra N Guha, Neil A Hanley, Karen Piper Hanley

Corresponding author: Karen Piper Hanley, University of Manchester

Review timeline:

Submission date:	12 April 2017
Editorial Decision:	19 May 2017
Additional Correspondence:	22 May 2017
Editorial Decision:	23 May 2017
Revision received:	17 August 2017
Editorial Decision:	05 September 2017
Revision received:	27 September 2017
Accepted:	02 October 2017

Transaction Report:

Editor: Céline Carret

1st Editorial Decision

19 May 2017

Thank you for the submission of your research manuscript to our editorial office. We have now received the enclosed reports on it.

As you will see, all three referees found the study of potential interest. However, while referee 1 is rather supportive, the other two reviewers raise serious concerns regarding the conclusiveness of the data and pinpoint several technical issues that preclude a solid interpretation of the experimental evidence provided. All of the reviewers would like to see more mechanism and are concerned about the role of Sox9 in cholangiocytes in the liver. We feel that to address it all would call for a considerable amount of additional experimentation that at this stage would go over the purpose of single revision within a 3-months deadline.

Therefore, given the amount of work likely to be required to address the criticisms raised, and the fact that EMBO Molecular Medicine can only invite revision of papers that receive enthusiastic support from a majority of referees at the initial review, I am afraid that we do not feel it would be productive to call for a revised version of your manuscript at this stage and therefore we cannot offer to publish it.

This said, because the study clearly has merits, we would have no objection to consider a new manuscript on the same topic if at some time in the near future you obtained data that would

considerably strengthen the message of the study and address the referees concerns in full. To be completely clear, however, I would like to stress that if you were to send a new manuscript this would be treated as a *new submission* rather than a revision and would be reviewed afresh, in particular with respect to the literature and the novelty of your findings at the time of resubmission. If you decide to follow this route, please make sure to nevertheless upload a letter of response to the referees' comments to help the process.

At this stage, though, I am sorry to have to disappoint you. I nevertheless hope, that the referee comments will be helpful in your continued work in this area and I thank you for considering EMBO Molecular Medicine.

***** Reviewer's comments *****

Referee #1 (Comments on Novelty/Model System):

The manuscript by Athwal et al provides some new insights in the field of liver fibrosis and cirrhosis. The authors have addressed relevant questions concerning liver fibrosis and progression to cirrhosis. Human data are provided with complete clinical data and some state of the art model techniques are presented. The manuscript text is very conclusive. To my feeling the cell culture part could be expanded.

Referee #1 (Remarks):

The manuscript by Athwal et al provides some new insights in the field of liver fibrosis and cirrhosis.
The whole manuscript is very conclusive and addresses a clinically relevant topic. The quality of the histological pictures is very good and representative.

I only have a few minor comments:

- it would be nice if the authors could provide a useful SOX9 grading system for liver biopsies that might help human pathologists to judge the impact of SOX9 in routine diagnostics.
- to my feeling, the fibrosis development using 1:3 CCl4 for 8 weeks is quite weak in the control animals. Did the authors see great variation throughout the whole liver? Do the authors have any explanation for that? My experience is that 1:3 CCl4 for 8 weeks leads to fully developed cirrhotic livers with prominent nodules. In the picture I see a nodule but the scra formation is quite weak.
- the authors should provide more information on other cell types in the liver (e.g. macrophage content, desmin staining and desmin/aSMA ratio to see the overall number of HSC in the livers).
- can you present any quantification on inflammatory cells in the established rodent models? It would be nice to see the amount of inflammation.
- Fig 2J: PSR in Fig 2I is significantly elevated in cnt mice after bdl. Can you explain why aSMA content is NOT significant? Please provide more data (e.g. stainings such as desmin) on that issue.
- please include H&E stains in all figures where histology is shown.
- the cell culture part should be strengthened by additional experiments supporting the authors' hypothesis (e.g. downstream signalling, western blot analysis...).
- the whole manuscript is very detailed. Unfortunately I don't find much information about exact animal numbers/experiment. How many times were the experiments repeated? Please specify for each experiment.

Referee #2 (Comments on Novelty/Model System):

Mice with HSC-specific deletion of Sox9 could be useful, rather than the use of whole body Sox9 KO.

Referee #2 (Remarks):

Athwal et al. investigated the role of the transcription factor Sex determining region Y box 9 (Sox9) in liver fibrosis in HCV patients and experimental models of liver fibrosis. They found that Sox9 in HSCs could be a predictor for liver fibrosis progression in HCV patients. In addition, they showed

that Sox9 is induced in HSCs upon activation and its loss (whole body knockout using Sox9^{fl/fl}, CreER^{+/-} mice) protects against fibrosis in two rodent models of fibrosis generated by CCl₄ ip injection for 8 weeks and BDL for 2 weeks. It was also shown that matrix stiffness similar to liver fibrosis (12KPa) induced expression and translocation of Sox9 into nucleus of HSCs and that this occurred along with the translocation of YAP and TAZ, suggesting the link among HSC activation, Sox9 induction and YAP/TAZ. This is a novel study with excellent data quality. However, overall the results are not enough to conclude the profibrotic role of Sox9 in HSCs. Specific comments are summarized below.

1. Studies from this group demonstrated in vitro that Sox9 was expressed in activated HSCs to promote ECM components, such as type I collagen and osteopontin. The current study expanded their previously published in vitro studies to human samples and Sox9 KO mice with liver fibrosis. Although these experiments could indicate Sox9 in HSCs is involved in the progression of liver fibrosis, it is not conclusive, since the pre-clinical experiment was performed in whole body Sox9 KO mice. Therefore, decreased liver fibrosis in Sox9 KO mice could be also due to Sox9 function in hepatocytes and/or cholangiocytes. Mice with HSC-specific deletion of Sox9 could be useful.
2. Fig 1B: the co-localization of SOX9 and HSC is not very clear.
3. Need some clarification for the numbers of patient biopsies in the Results section: "... (tissue remained from 152 biopsies for this purpose)" as well as "... (where tissue remained from 59 paired biopsies)." Where did these numbers, 152 biopsies and 59 paired biopsies come from, given that a total number of patients was 115?
4. Sox9 induction in HSCs cultured on stiff (12KPa) hydrogel is interesting. What is the significance of this induction? However, this induction would not necessarily indicate the pro-fibrotic role of Sox9 in HSCs. Does the loss of Sox9 in HSCs result in decreased fibrogenic phenotype compared to WT HSCs? How does Sox9 in HSCs promote fibrosis?

Referee #3 (Comments on Novelty/Model System):

Two models (CCl₄ and BDL) are very good models for investigating liver fibrosis. But, analyses using them were not appropriate. For instance, hepatic stellate cell-specific Cre is better than ROSA-CreER system.

Referee #3 (Remarks):

Based on the previous findings by the authors and others, the present hypothesis and application for the prediction of liver fibrosis depending on SOX9 expression is reasonable and important. Therefore, in the present study, the authors tried to explain the beneficial role of SOX9 for the prediction of liver fibrosis progression and the protective role of SOX9 against liver fibrosis. However, although the study provides some interesting data, it is not enough to draw a conclusion from the data presented by the study. Especially, the value of this study was damaged by several unclear data and uncertain mechanism although SOX9 depletion prevented liver fibrosis in CCl₄- and BDL-induced rodent models. First, the authors focused mainly in the role and expression of SOX9 in hepatic stellate cells (HSCs) although proliferating cholangiocytes in BDL showed more strong expression of SOX9 than HSCs. To exclude it, the authors must use HSC-specific SOX9 KO mice. Otherwise, the author tried to find good explanation to disregard the role of SOX9⁺ cholangiocytes. Second, YAP1-mediated SOX9 induction should be demonstrated not only in HSCs but also other types of cells such as hepatocyte and cholangiocytes. If the authors observe YAP1-mediated SOX9 induction only in HSCs, you should explain and discuss about it in the text. My concerns were raised as below.

Major concerns

- 1) Please check scale bar at Figure 1B, especially scales of two figures stained for SOX9 were same at the lower panels (IS6).
- 2) In the lower panel of Figure 1B, comparing with broad and strong staining of α -SMA around bile ducts at portal area in liver section of IS6, there were relatively small numbers of SOX9 positive cells observed except bile ductal cells although a lot of nuclei were detected in that area. What is the reason for unmatched staining in number and co-localization between α -SMA positive and SOX9-positive cells? Indeed, how could authors speculate that all SOX9⁺ cells with elongated nuclei are activated HSCs without double staining?

- 3) In Figure 1C, double staining with α -SMA and SOX9 was not clear to identify activated HSCs.
- 4) In Figure 1D, except cholangiocytes and some of hepatocytes, there was no exact correlation between EpCAM+ cells (maybe nascent hepatocytes to cholangiocytes) and SOX9+ cells (most of spindle cells) based on the sizes of nucleus and cytoplasm at IS6. To confirm this matter, the authors should show more background staining to see each line of cell membrane after SOX9 staining. Otherwise, the authors should add one more co-stained figures using SOX9 and another hepatocyte-specific antigen. Based on these unconvincing staining data, the remaining data of Figure 1 could not be evaluated.
- 5) In proportion to ECM accumulation, proliferation of bile ductal cells occurs in a variety of liver diseases such as cirrhotic liver. Thus, more numbers of cholangiocytes or bile ductular cells should be observed in IS4-6 compared with those of IS0 or IS1. In addition, these cells might be positive for SOX9. However, there was no difference of SOX9+ Biliary population among groups at Figure 1E.
- 6) In Figure 1G and Table 1S, numbers of two groups "non-progressors" and "progressors" are too small to gain statistical power. For instance, if the numbers of each group were larger, NI at first biopsy might become a significant factor even after the multivariate analysis. I recommend to assign more patients to each group or validate the findings in other cohorts.
- 7) Since HCV virus titer and the presence or absence of HCV treatment are significant factors in the progression of liver fibrosis, the authors are recommended to add the disease status of HCV infection at initial and follow-up biopsy in the table (e.g. HCV virus titer or if they received any treatment for HCV). And, they should include this factor in univariate and multivariate analysis.
- 8) In Figure 2A-C, according to many literatures and experimental studies, a lots of α -SMA-positive cells increase to produce ECM near proliferating bile ductal cells after BDL. However, in your study, SOX9-positive myofibroblasts or activated HSCs were not detected near bile ducts in BDL-induced liver fibrosis. How can the authors explain about this?
- 9) In legend of Figure 2B, the authors described that arrowheads indicated double positive cells for SOX9 and HNF4 α . However, there were no double positive cells in CCl4-treated liver.
- 10) In Figure 2C, to confirm isolated HSCs from olive oil- and CCl4-treated livers, more activated markers such as α -SMA, collagen or TGF- β 1 are needed.
- 11) As the authors suggested, if SOX9 expression in HSCs is important for the prediction of liver fibrosis, please show SOX9 positive HSCs in fibrosis septa of CCl4-treated liver in Figure 1E.
- 12) Similarly, in Figure 2H, protective effect of inactivating SOX9 was greater in peribiliary fibrosis model (BDL) than parenchymal model (CCl4 injection). I wonder if this phenomenon is due to SOX9 of cholangiocyte because SOX9 protein is constitutively expressed in cholangiocyte and BDL induces peribiliary fibrosis, followed by proliferation of bile ductal cells. Please explain and clarify this point in the discussion section or elsewhere.
- 13) In Figure 2, the authors should use a cell-type specific Cre (e.g. HSC-specific Cre such as PDGFR α -cre, AP2-cre or LRAT-cre rather than ROSA-Cre) if they want to explain the mechanism of SOX9-mediated HSC activation. How do the authors exclude off-target effect of SOX9 in cholangiocytes?
- 14) In Figure 3, like HSCs, what about Yap-mediated SOX9 regulation in cholangiocytes? If there is, what is the effects of it on liver fibrosis?
- 15) In Figure 3, the authors used acrylamide hydrogels of varying stiffness (4kPa or 12kPa) in vitro culture of HSCs to model changes in mechanical stiffness. Is it a well-validated model? Please explain and compare gene expression results such as α -SMA or Colla1 between 4kPa and 12kPa.

Minor concerns

- 1) In legend of Figure 2B, please check typo such as SOX-/HNF4 α + hepatocyte.
- 2) In legend of Figure S2, please check typo (e.g. Figureure 1 should be corrected into Figure 1).
- 3) In the Genotyping Mouse Strains section of supplementary methods page 9, Sox9 flox alleles should be corrected into Sox9 floxed alleles.

Additional Correspondence from Authors

22 May 2017

Thank you for your email and the comments. We have our hands up; we got this one wrong! We took out the data that the referees request to try and simplify the message e.g. additional animal models proving that SOX9's pro-fibrotic role does not reside in the cholangiocytes (or the hepatocytes) and detailed quantification of inflammatory cells in response to SOX9 loss. We would

certainly come back to the journal and (because the data are to hand) well within three months. Would it be possible to discuss with you how best to proceed?

2nd Editorial Decision

23 May 2017

Thank you for your e-mail.

I have discussed with my colleagues and in light of your message, we will be willing to give your manuscript another chance if you can address all comments from the referees and resubmit within the 3-months deadline. However, I'd like to stress that while I'll try to secure the same referees, I cannot guarantee it at this stage.

Nevertheless, I'll be happy to consider your revised article.

1st Revision - authors' response

17 August 2017

Referee #1 (Comments on Novelty/Model System):

The manuscript by Athwal et al provides some new insights in the field of liver fibrosis and cirrhosis. The authors have addressed relevant questions concerning liver fibrosis and progression to cirrhosis. Human data are provided with complete clinical data and some state of the art model techniques are presented. The manuscript text is very conclusive. To my feeling the cell culture part could be expanded.

Thank you for these comments. We have undertaken additional experiments.

Referee #1 (Remarks):

The manuscript by Athwal et al provides some new insights in the field of liver fibrosis and cirrhosis. The whole manuscript is very conclusive and addresses a clinically relevant topic. The quality of the histological pictures is very good and representative.

I only have a few minor comments:

- it would be nice if the authors could provide a useful SOX9 grading system for liver biopsies that might help human pathologists to judge the impact of SOX9 in routine diagnostics.

Thank you. It is rewarding that the reviewer, like us, immediately sees this potential. We have devised the SOX9 index in the manuscript and demonstrated important mechanistic roles for SOX9 in HSCs. It is important that we now disseminate these data as we believe they will be of timely interest to the broad fibrosis community (hence we have come to EMBO Molecular Medicine). We are now embarking on a new programme of work, as the reviewer suggests, into a SOX9 grading system. As the reviewer will appreciate, transition from a research to a routine clinical diagnostic setting requires many steps beyond those reported in original research papers and it is vital that we get those steps right. We are now undertaking this new work in combination with scrutinising other putative parameters of fibrosis prediction and progression. To share our thoughts out of interest, we feel that a combinatorial algorithmic approach may well offer the best utility for personalized medicine (e.g. SOX9 index, SOX9 downstream targets as liquid biopsy parameters, genomic risk factors such as the Cirrhosis Risk Score SNPs, elastography and data from the clinical history). Clearly, all this work is far beyond the scope of this current manuscript (but is exciting!). Nevertheless, the current work provides its mechanistic foundation.

- to my feeling, the fibrosis development using 1:3 CCl4 for 8 weeks is quite weak in the control animals. Did the authors see great variation throughout the whole liver? Do the authors have any explanation for that? My experience is that 1:3 CCl4 for 8 weeks leads to fully developed cirrhotic livers with prominent nodules. In the picture I see a nodule but the scra formation is quite weak.

We are 'live' to the potential issue of catching a misrepresentative region of liver both in our work and in the published literature. For this reason, all our data are quantified in a blinded process across the entire liver. We are reassured that the control livers are genuinely severely fibrotic, and that this is markedly ameliorated by loss of SOX9. For interest,

we provide another couple of lower magnification images indicating the extensive induction of fibrosis by CCl₄ in the control animals.

- the authors should provide more information on other cell types in the liver (e.g. macrophage content, desmin staining and desmin/aSMA ratio to see the overall number of HSC in the livers).
- can you present any quantification on inflammatory cells in the established rodent models? It would be nice to see the amount of inflammation.
- Fig 2J: PSR in Fig 2I is significantly elevated in cnt mice after bdl. Can you explain why aSMA content is NOT significant? Please provide more data (e.g. stainings such as desmin) on that issue.

Thank you for these helpful comments. We have incorporated extensive data on macrophages. In Figure 5 and Figure S12 we show FACS analysis of individual macrophage populations following CCl₄ induced fibrosis of control and Sox9-null animals. Interestingly, the data show monocyte recruitment is unimpaired following loss of SOX9 (i.e. lessened fibrosis) but that macrophage maturation is curtailed. Sadly, we have found data on desmin difficult to interpret as it is expressed in both quiescent and activated HSCs. To get around this, as requested, in Figure S5 we provide additional staining for COL1, which is decreased in response to SOX9 loss (these data are also backed up by our previous publications and Figure S7).

- please include H&E stains in all figures where histology is shown.

H&E staining for histology figures has been included in Figure S4.

- the cell culture part should be strengthened by additional experiments supporting the authors' hypothesis (e.g. downstream signalling, western blot analysis...).

We were not quite sure of the specific request here? We have already published several manuscripts using cell culture models of liver fibrosis to provide detailed mechanistic insight into the role of SOX9 (Hanley et al, JBC, 2008; Pritchett et al, Hepatology, 2012; Pritchett et al, PlosOne, 2013, Martin et al, Nat. Comms, 2016). These publications really pressed the case for the current in vivo study and consequently we believe our new findings will be high impact for the field. Of note, we have included additional Western data in Figure S7 from in vitro activated HSCs following SOX9 loss compared to control cells.

- the whole manuscript is very detailed. Unfortunately I don't find much information about exact animal numbers/experiment. How many time were the experiments repeated? Please specify for each experiment.

Apologies, to meet the word count limits in figure legends we originally placed the animal numbers for each experiment in the Materials and Methods. We have now modified this with additional detail included in the Appendix's Supplementary Methods. Of note, animals used in each experimental protocol (at least n=5) were in line with our power calculations to allow the attainment of significant results for each fibrosis model.

Referee #2 (Comments on Novelty/Model System):

Mice with HSC-specific deletion of Sox9 could be useful, rather than the use of whole body Sox9 KO.

Thank you for this comment, which is also picked up on by Referee 3. We have addressed this—detailed below.

Referee #2 (Remarks):

Athwal et al. investigated the role of the transcription factor Sex determining region Y box 9 (Sox9) in liver fibrosis in HCV patients and experimental models of liver fibrosis. They found that Sox9 in HSCs could be a predictor for liver fibrosis progression in HCV patients. In addition, they showed that Sox9 is induced in HSCs upon activation and its loss (whole body knockout using Sox9^{fl/fl}, CreER^{+/-} mice) protects against fibrosis in two rodent models of fibrosis generated by CCl₄ ip injection for 8 weeks and BDL for 2 weeks. It was also shown that matrix stiffness similar to liver fibrosis (12KPa) induced expression and translocation of Sox9 into nucleus of HSCs and that this occurred along with the translocation of YAP and TAZ, suggesting the link among HSC activation, Sox9 induction and YAP/TAZ. This is a novel study with excellent data quality. However, overall the results are not enough to conclude the profibrotic role of Sox9 in HSCs.

We thank the reviewer for these very kind comments. We appreciate the final conclusion and hence we have included additional data, which, on reflection, should perhaps have been included in the original submission.

Specific comments are summarized below.

1. Studies from this group demonstrated in vitro that Sox9 was expressed in activated HSCs to promote ECM components, such as type I collagen and osteopontin. The current study expanded their previously published in vitro studies to human samples and Sox9 KO mice with liver fibrosis. Although these experiments could indicate Sox9 in HSCs is involved in the progression of liver fibrosis, it is not conclusive, since the pre-clinical experiment was performed in whole body Sox9 KO mice. Therefore, decreased liver fibrosis in Sox9 KO mice could be also due to Sox9 function in hepatocytes and/or cholangiocytes. Mice with HSC-specific deletion of Sox9 could be useful.

Thank you. We agree. Apologies for the long answer, but this is an important point and chimes with comments from Reviewer 3.

The reviewer correctly questions whether SOX9 in hepatocytes or cholangiocytes might contribute to the transcription factor's profibrotic role. We directly addressed this hypothesis in vivo (and should have included the data previously!) by inactivating SOX9 in both hepatocyte and cholangiocyte lineages using AlbCre (both lineages develop from Alb⁺ progenitors). SOX9 is lost in hepatocytes and cholangiocytes but not in HSCs. Fibrosis was unaltered from controls (Figure 4C-E and Appendix Figure S11), clearly addressing the reviewer's concern and indicating that SOX9's profibrotic function does not reside in hepatocytes or cholangiocytes but in the HSCs.

As part of this work, we include data in tissue sections using IHC and RNAScope in situ hybridisation for protein and transcript detection showing SOX9 in discrete α Sma positive cells (HSCs) within the scar (Figure 4A, B and Appendix Figure S8). To further demonstrate retention of the SOX9 gene in the HSC lineage, in vitro activation of HSCs from Sox9^{fl/fl};AlbCre^{+/-} livers showed normal induction of SOX9 protein (Figure 4F, G and Appendix Figure S9). As the reviewer points out, these data concord with our previous in vitro work (Hanley et al, JBC, 2008; Pritchett et al, Hepatology, 2012; Pritchett et al, PlosOne, 2013, Martin et al, Nat. Comms, 2016).

The comment about HSC-specific deletion is welcome. Importantly, no suitable current so-called HSC-specific Cre models are suitable as they are non-inducible and compromised by severe defects and lethality due to SOX9 loss in other cell lineages during embryogenesis (e.g. neural crest cells). Because this is such an important point that frequently gets trivialised we have included new text on this point in the manuscript:

'Given the critical developmental role for SOX9 in multiple tissues (Pritchett et al, 2011), it was not possible to knock out SOX9 in HSCs using the currently published PdgfrB, AP2 and LRAT Cre models (Henderson et al, 2013; Mederacke et al, 2013; Moran-Salvador et al, 2013). PdgfrB expression overlaps SOX9 in neural crest cell (NCC) populations responsible for heart development (Akiyama et al, 2004; Smith & Tallquist, 2010; Van den Akker et al, 2008). In keeping with this, >80% of our Sox9^{fl/fl};PdgfrBCre⁺ mice die during embryogenesis/birth. AP2 also has a major role in NCCs (and chondrogenesis) (Pritchett et al, 2011; Wenke & Bosserhoff, 2010). LRAT expression

overlaps SOX9 during development (e.g. in liver, lung, pancreas (Batten et al, 2004; Jennings et al, 2013; Pritchett et al, 2011)). Thus, none of these Cre drivers could be suitable for HSC-specific SOX9 inactivation.'

2. Fig 1B: the co-localization of SOX9 and HSC is not very clear.

Thank you. The co-localisation data for SOX9/ α SMA to support Figure 1B was included in Figure 1C. We have not added new data (Figures 4 & 5) and RNAscope in situ hybridisation (ISH; Figures 2 and 4).

3. Need some clarification for the numbers of patient biopsies in the Results section: "... (tissue remained from 152 biopsies for this purpose)" as well as ".... (where tissue remained from 59 paired biopsies)." Where did these numbers, 152 biopsies and 59 paired biopsies come from, given that a total number of patients was 115?

Apologies for any confusion. To help clarify the patient numbers we added a flow diagram to explain the patient numbers for each stage of analysis (Figure EV1).

4. Sox9 induction in HSCs cultured on stiff (12KPa) hydrogel is interesting. What is the significance of this induction? However, this induction would not necessarily indicate the pro-fibrotic role of Sox9 in HSCs. Does the loss of Sox9 in HSCs result in decreased fibrogenic phenotype compared to WT HSCs? How does Sox9 in HSCs promote fibrosis?

Thank you. We agree that this is interesting. Culturing HSCs on stiff hydrogels provides a mechanistic basis for SOX9 induction in response to YAP, as described in this study (Figure 6) and supported by our data in Martin et al, Nature Comms, 2016. The data in this manuscript clearly indicate a profibrotic role for SOX9, now resolved to HSCs, and build on our previous in vitro work specifically showing that loss of SOX9 in HSCs reduces the profibrotic phenotype (Hanley et al, JBC, 2008; Pritchett et al, Hepatology, 2012; Pritchett et al, PlosOne, 2013, Martin et al, Nat. Comms, 2016). To help further with this point, we have also included a new supplementary image in Figure S7.

Referee #3 (Comments on Novelty/Model System):

Two models (CCl4 and BDL) are very good models for investigating liver fibrosis. But, analyses using them were not appropriate. For instance, hepatic stellate cell-specific Cre is better than ROSACreER system.

Thank you. We have addressed the mechanistic point underlying this comment in an additional animal model and explained that so-called 'HSC-specific' Cre is not possible for the deletion of SOX9. This is explained in detail in comments to Reviewer 2.

Referee #3 (Remarks):

Based on the previous findings by the authors and others, the present hypothesis and application for the prediction of liver fibrosis depending on SOX9 expression is reasonable and important. Therefore, in the present study, the authors tried to explain the beneficial role of SOX9 for the prediction of liver fibrosis progression and the protective role of SOX9 against liver fibrosis. However, although the study provides some interesting data, it is not enough to draw a conclusion from the data presented by the study. Especially, the value of this study was damaged by several unclear data and uncertain mechanism although SOX9 depletion prevented liver fibrosis in CCl4- and BDL-induced rodent models. First, the authors focused mainly in the role and expression of SOX9 in hepatic stellate cells (HSCs) although proliferating cholangiocytes in BDL showed more strong expression of SOX9 than HSCs. To exclude it, the authors must use HSC-specific SOX9 KO mice.

Otherwise, the author tried to find good explanation to disregard the role of SOX9+ cholangiocytes. Second, YAP1-mediated SOX9 induction should be demonstrated not only in HSCs but also other types of cells such as hepatocyte and cholangiocytes. If the authors observe YAP1-mediated SOX9

induction only in HSCs, you should explain and discuss about it in the text. My concerns were raised as below.

Please see comments to Reviewer 2. We are grateful that the reviewer recognises that 'SOX9 depletion prevented liver fibrosis' (above) in the two 'very good models for investigating liver fibrosis' (the earlier comment). This is the crux of the paper and it is rewarding that the reviewer recognises this.

The reviewer specifically comments on the cholangiocytes. We have now excluded any significant profibrotic role for SOX9 in cholangiocytes by deleting the transcription factor in this cell lineage. (Similarly, we have excluded a profibrotic role for SOX9 in the hepatocytes). As the only other cell-type in the liver containing SOX9 is the activated HSC this proves that SOX9's profibrotic function resides in this latter cell-type. While, theoretically, HSC-specific SOX9 KO mice would be another way to approach this, we have explained why this is not possible due to lethality in comments to Reviewer 2. We have added new text explaining this point in the manuscript. Having excluded cholangiocytes, the comment on YAP induction of SOX9 in cholangiocytes becomes obsolete in the context of liver fibrosis.

Major concerns

1) Please check scale bar at Figure 1B, especially scales of two figures stained for SOX9 were same at the lower panels (IS6).

Thank you for highlighting this. We had missed this. We have updated this oversight.

2) In the lower panel of Figure 1B, comparing with broad and strong staining of α -SMA around bile ducts at portal area in liver section of IS6, there were relatively small numbers of SOX9 positive cells observed except bile ductal cells although a lot of nuclei were detected in that area. What is the reason for unmatched staining in number and co-localization between α -SMA positive and SOX9-positive cells? Indeed, how could authors speculate that all SOX9+ cells with elongated nuclei are activated HSCs without double staining?

Please see below which addresses this comment (we have added new data).

3) In Figure 1C, double staining with α -SMA and SOX9 was not clear to identify activated HSCs.

The colocalisation data for SOX9/ α SMA in Figure 1C supports the serial section data in Figure 1B. We now back-up these data with additional in vivo experiments (Figures 4 & 5) and RNAScope in situ hybridisation (ISH; Figures 2 and 4).

4) In Figure 1D, except cholangiocytes and some of hepatocytes, there was no exact correlation between EpCAM+ cells (maybe nascent hepatocytes to cholangiocytes) and SOX9+ cells (most of spindle cells) based on the sizes of nucleus and cytoplasm at IS6. To confirm this matter, the authors should show more background staining to see each line of cell membrane after SOX9 staining. Otherwise, the authors should add one more co-stained figures using SOX9 and another hepatocyte-specific antigen. Based on these unconvincing staining data, the remaining data of Figure 1 could not be evaluated.

This comment is at odds with those of the other reviewers. We do not understand the point that is being made? It seems unusual and unhelpful to request 'more background staining'. The SOX9 data in this figure (and the rest of the manuscript) is very clear. The data on EpCAM are an interesting observation but not pivotal to the paper's mechanistic thrust. If the editorial team felt strongly, they could be removed from the manuscript. We defer to editorial opinion.

5) In proportion to ECM accumulation, proliferation of bile ductal cells occurs in a variety of liver diseases such as cirrhotic liver. Thus, more numbers of cholangiocytes or bile ductular cells should be observed in IS4-6 compared with those of IS0 or IS1. In addition, these cells might be positive for SOX9. However, there was no difference of SOX9+ Biliary population among groups at Figure 1E.

HCV infection induces mostly parenchymal liver fibrosis, which at IS4-6 is obvious. It seems unusual and unhelpful to prejudge what data should look like by requiring 'more numbers of

cholangiocytes'. Fibrotic diseases are complex involving varying degrees of ductal proliferation, ductopenia, tissue destruction, distortion, cellular alterations and senescence associated with all cell types. Because of this, the quantification of ducts is not commonly used clinically to grade severity.

6) In Figure 1G and Table 1S, numbers of two groups "non-progressors" and "progressors" are too small to gain statistical power. For instance, if the numbers of each group were larger, NI at first biopsy might become a significant factor even after the multivariate analysis. I recommend to assign more patients to each group or validate the findings in other cohorts.

At odds with the other reviewers, we think this reviewer is somewhat missing the point here. SOX9 data are significant through multiple statistical modelling, while NI is not. We do not understand how this can be interpreted as a need for larger groups? (Interestingly, while including NI here is a useful comparator given its previous attention, several studies in some of the largest international cohorts have demonstrated that biopsy NI evidence is not conclusively associated with liver fibrosis progression, if at all (Poynard T, Lancet, 1997; Ghany, Gastro, 2003; Zeremski, J Infect Dis, 2016).)

7) Since HCV virus titer and the presence or absence of HCV treatment are significant factors in the progression of liver fibrosis, the authors are recommended to add the disease status of HCV infection at initial and follow-up biopsy in the table (e.g. HCV virus titer or if they received any treatment for HCV). And, they should include this factor in univariate and multivariate analysis.

There is inconclusive evidence to support viral load as a single predicative measure of fibrosis (Heller & Seeff, Hepatology, 2005). Thus, it would be inappropriate and unhelpful to include it here (compared to proven clinical patient demographics). Thank for raising the point about HCV eradication treatment. It is unnecessary to include this factor in the analysis as none of the patients were treated during the time course of the tissue biopsies. We have made this point in the text.

8) In Figure 2A-C, according to many literatures and experimental studies, a lots of α -SMA-positive cells increase to produce ECM near proliferating bile ductal cells after BDL. However, in your study, SOX9-positive myofibroblasts or activated HSCs were not detected near bile ducts in BDL-induced liver fibrosis. How can the authors explain about this?

Periductal fibrosis was present post-BDL in our study (as one would expect) so we don't entirely understand what the reviewer is referring to? We trust that the additional IHC and in situ hybridisation data (Figures 2 and 4) help in this regard.

9) In legend of Figure 2B, the authors described that arrowheads indicated double positive cells for SOX9 and HNF4 α . However, there were no double positive cells in CCl4-treated liver.

Thank you for spotting this. Apologies, this was suboptimally written in the figure legend—there were also arrowheads showing SOX9-positive CK19 negative cells. We have addressed this (the individual channels are included for clarity in Figure S1 as referred to in the text of the manuscript).

10) In Figure 2C, to confirm isolated HSCs from olive oil- and CCl4-treated livers, more activated markers such as α -SMA, collagen or TGF- β 1 are needed.

Thank you. This chimes with earlier comments and we have included these data in Figure S2.

11) As the authors suggested, if SOX9 expression in HSCs is important for the prediction of liver fibrosis, please show SOX9 positive HSCs in fibrosis septa of CCl4-treated liver in Figure 1E.

Thank you. In line with comments from reviewer 1, we have now included COL1 staining (Figure S5) and dual localisation with SOX9/SMA using immuno and in situ hybridisation using RNAscope (Figure 2C).

12) Similarly, in Figure 2H, protective effect of inactivating SOX9 was greater in peribiliary fibrosis model (BDL) than parenchymal model (CCl4 injection). I wonder if this phenomenon is due to

SOX9 of cholangiocyte because SOX9 protein is constitutively expressed in cholangiocyte and BDL induces peribiliary fibrosis, followed by proliferation of bile ductal cells. Please explain and clarify this point in the discussion section or elsewhere.

We have now provided data to conclusively exclude a profibrotic role for SOX9 in cholangiocytes which closes this comment.

13) In Figure 2, the authors should use a cell-type specific Cre (e.g. HSC-specific Cre such as PDGFR α -cre, AP2-cre or LRAT-cre rather than ROSA-Cre) if they want to explain the mechanism of SOX9-mediated HSC activation. How do the authors exclude off-target effect of SOX9 in cholangiocytes?

Please see our reply to this in the comments to reviewer 2. We have added comprehensive new data (Figure 4 and Appendix Figures S8-S11).

14) In Figure 3, like HSCs, what about Yap-mediated SOX9 regulation in cholangiocytes? If there is, what is the effects of it on liver fibrosis?

As earlier, this point now becomes obsolete having excluded a profibrotic role for SOX9 in cholangiocytes. Beyond liver fibrosis (i.e. beyond the focus of our paper), we refer the reviewer to the work of others who have published on the role of YAP in cholangiocytes (Zhang et al, Dev. Cell, 2010; Yimlamai et al, Cell, 2014).

15) In Figure 3, the authors used acrylamide hydrogels of varying stiffness (4kPa or 12kPa) in vitro culture of HSCs to model changes in mechanical stiffness. Is it a well-validated model? Please explain and compare gene expression results such as α -SMA or Col1a1 between 4kPa and 12kPa.

This is an extensively used and well validated model to investigate mechanosignalling in many cell types, including HSCs (Wells et al, Gastro, 2005; Olsen et al, Am. J. Physiol-Gastro. & Liver Physiol, 2011). We back up our data here in vivo (Figure 6D-F) and by our recent publication (Martin et al, Nat Comms, 2016).

Minor concerns

- 1) In legend of Figure 2B, please check typo such as SOX $^{-/-}$ /HNF4 α $^{+/+}$ hepatocyte.
- 2) In legend of Figure S2, please check typo (e.g. Figureure 1 should be corrected into Figure 1).
- 3) In the Genotyping Mouse Strains section of supplementary methods page 9, Sox9 flox alleles should be corrected into Sox9 floxed alleles.

Thank you – we have made these corrections.

3rd Editorial Decision

05 September 2017

Thank you for the submission of your revised manuscript to EMBO Molecular Medicine. We have now received the enclosed reports from referee 2 who reviewed it before and as unfortunately, the other referees were not available, we asked one of our board member for an additional editorial advice. The reviewing process is now completed and I am pleased to inform you that we will be able to accept your manuscript pending the following final amendments:

1) Please see the comments from our advisor and address the minor issues noted. Please provide a point by point rebuttal letter as well.

2) Source Data:

We now encourage the publication of source data, particularly for electrophoretic gels, blots, but also microscopy images with the aim of making primary data more accessible and transparent to the reader. Would you be willing to provide a PDF file per figure that contains the original, uncropped and unprocessed scans of all or key gels used in the figure? The PDF files should be labeled with the appropriate figure/panel number (1 file/figure), and should have molecular weight markers; further annotation may be useful but is not essential. The PDF files will be published online with the article

as supplementary "Source Data" files. If you have any questions regarding this just contact me.

Please submit your revised manuscript within two weeks. I look forward to seeing a revised form of your manuscript as soon as possible.

***** Reviewer's comments *****

Referee #2 (Comments on Novelty/Model System for Author):

The authors significantly improve the quality of the manuscript.

Referee #2 (Remarks for Author):

The authors sufficiently addressed my all four comments.

Editorial adviser:

"[...] the human analyses look to me quite strong and convincing. The authors have analyzed a substantial cohort of human samples and the analysis reads quite solid and straightforward. The association of fibrosis to HSC/hepatocyte vs. biliary expression looks quite solid to me. And I don't think any of the analyses would change if a larger cohort of samples was analyzed (just one minor issue to Fig. 1G: There appears to be a calculation mistake in the image: When adding up the biliary and the HSC/Hep bars in the progressor group, this will yield a larger bar than the „total" bar, i.e., there is either a graphical error or a calculation error, but, as presented, there must be a mistake in the „progressor" group).

Other than that, I believe that the authors have performed very solid revisions and they have submitted a substantially advanced revised manuscript that, I believe, will make a strong contribution to the journal. Notably, the amount of extra conditional mutagenesis to assign the experimental phenotype to the HSC compartment is quite impressive and convincing. The negative experiment using Alb-Cre as driver is pretty compelling.

Taken together, I have come to conclude that this MS deserves to be further pursued."

2nd Revision - authors' response

27 September 2017

Referee #2 (Comments on Novelty/Model System for Author):

The authors significantly improve the quality of the manuscript.

Referee #2 (Remarks for Author):

The authors sufficiently addressed my all four comments.

The referee has been very helpful and we hope the manuscript is stronger – thank you.

Editorial adviser:

"[...] the human analyses look to me quite strong and convincing. The authors have analyzed a substantial cohort of human samples and the analysis reads quite solid and straightforward. The association of fibrosis to HSC/hepatocyte vs. biliary expression looks quite solid to me. And I don't think any of the analyses would change if a larger cohort of samples was analyzed (just one minor issue to Fig. 1G: There appears to be a calculation mistake in the image: When adding up the biliary and the HSC/Hep bars in the progressor group, this will yield a larger bar than the "total" bar, i.e., there is either a graphical error or a calculation error, but, as presented, there must be a mistake in the "progressor" group).

Thank you for these comments and apologies, there was an error in analysis of one point in the progressor group. This has been corrected in Fig. 1G.

Other than that, I believe that the authors have performed very solid revisions and they have submitted a substantially advanced revised manuscript that, I believe, will make a strong contribution to the journal. Notably, the amount of extra conditional mutagenesis to assign the experimental phenotype to the HSC compartment is quite impressive and convincing. The negative experiment using Alb-Cre as driver is pretty compelling.

Taken together, I have come to conclude that this MS deserves to be further pursued."

Thank you for your comments and decision.

Corresponding Author Name: Karen Piper Hanley

Manuscript Number: EMM-2017-07860-V2